# A mechanism to initiate emergency type 2 myelopoiesis

Alexandre Fagnan[1], Cristina Di Genua[1], Yiran Meng[1], Roy Drissen[1], Zishan Zhang[1], Bowen Zhang[1], Padraic G. Fallon[2], Vassilis Pachnis[3], Erika J. Mancini[4], Fränze Progatzky[3,5] & Claus Nerlov[1✉]

Immune responses to parasite infection involve the increased production of basophils and eosinophils. These two myeloid cell types have key roles in type 2 anti-parasite immunity[1] and rely on GATA family transcription factors for their specification[2,3]. The first committed step in basophil and eosinophil production is generation of basophil–eosinophil–mast cell progenitors (BEMPs) from oligopotent erythroid-primed multipotent progenitors (EMPPs). However, it is not well established how immune responses act on progenitors to initiate type 2 myelopoiesis. Here we show that infection with the helminth *Heligmosomoides polygyrus* increases EMPP commitment to myeloid fate at the expense of erythropoiesis. Upon infection with *H. polygyrus*, the IL-33 alarmin accumulated in the bone marrow, causing EMPPs to upregulate the GATA co-factor LMO4 and preferentially differentiate into myeloid cells. LMO4 was sufficient to instruct myeloid fate in EMPPs by interacting with GATA2, displacing the FOG1 co-factor and redistributing GATA binding from megakaryocyte–erythroid-specific to basophil, eosinophil and mast cell (BEM)-specific chromatin. Accordingly, mice carrying a GATA2 mutation that selectively impairs the LMO4–GATA2 interaction were deficient in GATA factor allocation to BEM chromatin, myeloid lineage commitment, basophil and eosinophil production, and parasite control. This identifies LMO4 as an IL-33-regulated master regulator of type 2 myelopoiesis, and transcription factor reallocation as a mechanism of lineage commitment.

BEMs are key effector cells of the type 2 immune response, originating in the bone marrow and acting in barrier tissues to detect and fight parasites[1]. Dysregulated type 2 immunity is commonly associated with allergic inflammation[1], and excessive production of mast cells[4] or eosinophils[5] can lead to severe morbidity due to toxicity and inflammatory properties of the antimicrobial proteins produced by these cell types. Therefore, understanding the cellular pathways and molecular mechanism through which type 2 myeloid cells arise is critical if we are to understand how type 2 inflammation is regulated, and develop targeted strategies to correct imbalanced myeloid cell production.

Upon helminth infection and allergic inflammation, alarmins such as thymic stromal lymphopoietin, interleukin-25 (IL-25) and IL-33, are released from injured cells, inducing and enhancing the recruitment and expansion of type 2 immune cells[1]. Although direct action of type 2 inflammatory signals on bone marrow-resident-restricted basophil[6] or eosinophil[7] progenitors is well established, whether such signalling initiates type 2 myelopoiesis by promoting commitment to a BEM fate remains to be understood.

Recent studies have shown that BEMs are specified via a cellular pathway distinct from that generating neutrophils and monocytes in both mice and humans. In this pathway, the first myeloid-committed step is the generation of BEMPs from EMPPs, which in addition to BEMs generate erythrocytes and platelets via pre-megakaryocyte–erythroid progenitors (preMegEs)[8–11]. GATA2 has been shown to have a key role in the specification of BEMs[12]. However, *Gata2* is also critical for megakaryocyte and erythroid lineage development[13] and expressed at a similar level in BEMP and preMegEs[8]. Furthermore, FOG1, a GATA co-factor identified as interacting with the zinc-finger 1 (ZnF1) of GATA1 (ref. 14), is essential for the development of embryonic[15] and definitive[16] megakaryocyte and erythroid lineages, and ectopic expression of FOG1 has been shown to block both eosinophil[17] and mast cell[18] differentiation, consistent with GATA2 interactors, rather than the GATA2 expression level, having a key role in the BEMP versus preMegE lineage choice. However, a GATA2 co-factor that promotes BEM lineage commitment is yet to be identified.

Here we show that initiation of a type 2 immune response in mice by helminth infection induces myeloid bias in EMPPs, concomitant with an increase in bone marrow IL-33. IL-33 directly induced *Lmo4* expression in EMPPs, leading to GATA factor reallocation to BEM-specific chromatin through FOG1 displacement, EMPP myeloid bias and increased basophil and eosinophil formation. Conversely, a GATA2 mutation that selectively blocked the LMO4–GATA2 interaction had the opposite effect. Together, these results identify LMO4 as an IL-33-regulated

[1]MRC Molecular Haematology Unit, MRC Weatherall Institute of Molecular Medicine, University of Oxford, John Radcliffe Hospital, Oxford, UK. [2]School of Medicine, Trinity College Dublin, Dublin, Ireland. [3]Nervous System Development and Homeostasis Laboratory, Francis Crick Institute, London, UK. [4]School of Life Sciences and Sussex Drug Discovery Centre, University of Sussex, Brighton, UK. [5]Kennedy Institute of Rheumatology, Oxford, UK. ✉e-mail: claus.nerlov@imm.ox.ac.uk

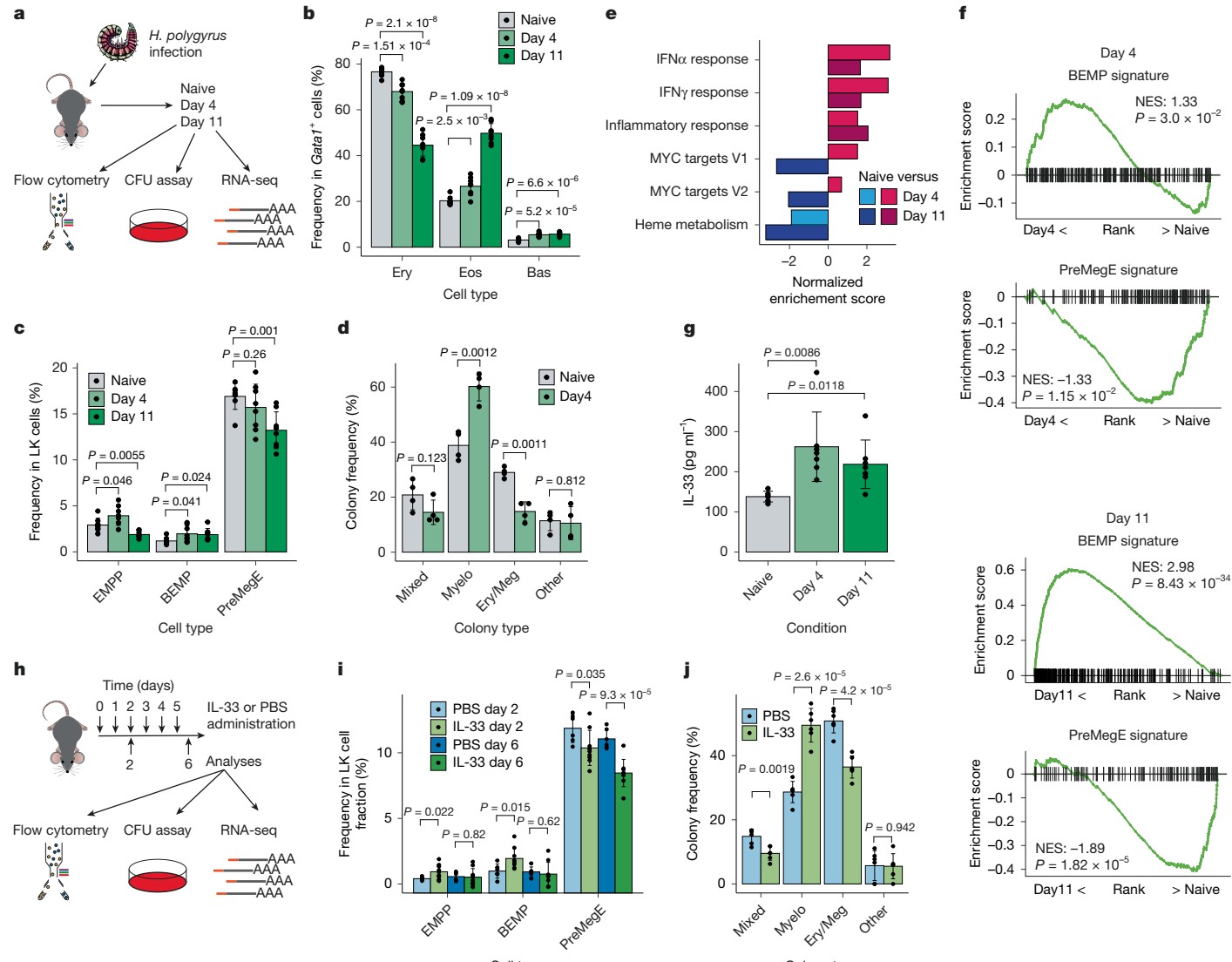

**Fig. 1 | *H. polygyrus* infection promotes EMPP myeloid bias. a**, Experimental workflow of *H. polygyrus* infection. Mice were infected for 4 or 11 days and compared with non-infected (naive) mice from the same cohort. CFU, colony-forming unit. **b**, Frequency of erythrocyte (Ery), eosinophil (Eos) and basophil (Bas) within the *Gata1*-expressing cell fraction in the bone marrow of mice days 4 and 11 post-infection as outlined in **a**. $n = 8$ biological replicates, 4 independent experiments. **c**, Frequency of EMPPs, BEMPs and preMegEs within the bone marrow LK cell fraction of mice from **a**. **d**, Quantification of clonal EMPP fates from **c**. Ery/Meg, Ery and/or Meg colonies; Myelo, mast cell (Mas) and/or Eos; mixed, Myelo and Ery/Meg; other, monocyte (Mon)/neutrophil (Neu). **e**, GSEA analysis showing enrichment of MSigDB hallmark gene sets ($P < 0.05$) comparing naive and day 4 or day 11 infected EMPPs as indicated using RNA-seq ($n = 4$

per condition). The normalized enrichment score (NES) is shown for each comparison. **f**, GSEA comparing expression of the BEMP (top) and preMegE (bottom) gene expression signatures between EMPPs isolated from naive mice to day 4 or day 11 infected mice. For RNA-seq, $n = 4$ per condition. NES and *P* values are shown for each comparison. **g**, The concentration of IL-33 was measured by ELISA in bone marrow supernatant of naive mice ($n = 6$) or mice infected with *H. polygyrus* for 4 or 11 days ($n = 7$ for each). **h**, Experimental workflow for analysis of mice treated with IL-33 or vehicle (PBS). **i**, Frequency of EMPPs, BEMPs and preMegEs in bone marrow LK cells of mice treated as outlined in **h**. PBS day 2, $n = 6$; PBS day 6, $n = 6$; IL-33 day 2, $n = 9$; and IL-33 day 6, $n = 9$ biological replicates; 3 independent experiments. **j**, Frequency of clonal EMPP fates from Extended Data Fig. 1f,g; six independent experiments.

inducer of type 2 myelopoiesis, and GATA factor reallocation as its mechanism of action.

## Helminth infection induces myeloid bias

To determine the effect of helminth infection on type 2 myelopoiesis, we infected mice with *H. polygyrus* and analysed the bone marrow by flow cytometry at 4 and 11 days post-infection (Fig. 1a and Extended Data Fig. 1a,b). As previously reported[6], we observed a progressive increase in basophils and eosinophils, concomitant with a decreased frequency of erythroid cells in the bone marrow of infected mice (Fig. 1b). In parallel, we observed an increased BEMP and decreased preMegE frequency

within Lin⁻Sca-1⁻Kit⁺ (LK) progenitors (Fig. 1c and Extended Data Fig. 1c,d), indicating that increased myeloid output was initiated by altered EMPP lineage bias. To test this hypothesis, we analysed EMPPs isolated 4 days post-infection in colony-forming assays, and analysed the resulting colonies by flow cytometry (Extended Data Fig. 1e). This showed increased myeloid and decreased erythroid–megakaryocyte (Ery–Meg) output from EMPP isolated from infected mice compared with EMPPs from naive mice (Fig. 1d and Extended Data Fig. 1f,g). To address the underlying molecular mechanism, we compared EMPP purified from infected mice to naive EMPPs using RNA sequencing (RNA-seq; Supplementary Tables 1 and 2). Gene set enrichment analysis (GSEA) showed upregulation of inflammatory signatures both day

4 and day 11 post-infection (Fig. 1e). Furthermore, using RNA-seq of EMPPs as well as the downstream BEMP and preMegE populations (Extended Data Fig. 2a) to identify BEMP-specific and preMegE-specific gene expression signatures (Extended Data Fig. 2b and Supplementary Table 3), we found that the BEMP signature was upregulated in EMPPs isolated from infected compared with naive mice, with a corresponding downregulation of the preMegE signature (Fig. 1f). Therefore, *H. polygyrus* infection generates an inflammatory environment within the bone marrow that induces molecular and functional myeloid bias in oligopotent myelo-erythroid progenitors.

## IL-33 induces *Lmo4* expression

Alarmins are released rapidly in response to parasite-induced tissue damage and have been shown to induce basophil[19] and eosinophil[20] expansion in the bone marrow. We therefore investigated whether alarmins could contribute to the observed myeloid bias in EMPPs after *H. polygyrus* infection. The genes encoding the IL-25 (*Il17ra*) and IL-33 (*Il1rl1*) receptors, but not the thymic stromal lymphopoietin receptor (*Crlf2*), were robustly expressed in EMPPs (Extended Data Fig. 2c). Furthermore, *Il1rl1*, but not *Il17ra*, expression was upregulated in BEMPs, and the concentration of IL-33 was higher in the bone marrow fluid from infected mice than in naive mice (Fig. 1g), consistent with a role for IL-33 in the induction of type 2 myelopoiesis. To test this possibility, mice received daily injections of IL-33 (or vehicle; Fig. 1h). This resulted in an increased BEMP and decreased preMegE frequency after 2 days of treatment, with a further decrease in frequency of preMegEs after 6 days (Fig. 1i and Extended Data Fig. 2d). In line with previous reports[7,20], treatment with IL-33 induced a large increase of eosinophils in the bone marrow after 6 days at the expense of erythroid cells (Extended Data Fig. 2e,f). Furthermore, a colony-forming assay of EMPPs from mice IL-33 treated for 2 days showed increased myeloid colony frequency and decreased frequency of Ery–Meg colonies compared with control mice (Fig. 1j and Extended Data Fig. 2g). Accordingly, RNA-seq and GSEA of EMPPs isolated from mice after 2 days of IL-33 treatment showed increased BEMP signature and decreased preMegE signature expression (Extended Data Fig. 2h). IL-33 therefore instructs molecularly and functional myeloid bias in EMPPs, similarly to what is observed after *H. polygyrus* infection.

## *Lmo4* instructs myeloid fate in EMPPs

To identify putative molecular mechanisms underlying myeloid bias induced by the type 2 immune response, we performed differential transcription factor gene expression analysis of EMPPs after *H. polygyrus* infection or IL-33 treatment (Extended Data Fig. 3a–c), and identified transcription factors showing the same regulation in both conditions (Fig. 2a). This identified *Klf1*, *Zfpm1* and *Tcerg1* as preMegE-specific transcription factors downregulated in both scenarios. Conversely, four transcription factors (*Jup*, *Zbtb33*, *Runx1* and *Lmo4*) were upregulated in both conditions, of which *Runx1* and *Lmo4* were more highly expressed in BEMPs compared with preMegEs (Fig. 2a and Supplementary Table 4). Of these, *Lmo4* was of particular interest, as it is a GATA2 co-factor[21], and is highly expressed in basophil and eosinophil progenitors[11,22], consistent with a role in specification of these lineages. However, although *lmo4* knockdown in zebrafish embryos impaired definitive haematopoiesis[23], the role of *Lmo4* in lineage specification remains to be understood.

We therefore expressed *Lmo4* by lentiviral delivery in EMPPs (Fig. 2b), using EMPPs isolated from *Gata1*–eGFP mice[8] to facilitate both EMPP isolation and subsequent lineage readout, and flow cytometry to quantify their lineage output at the single-cell level after 10 days. *Lmo4* expression led to an increase in the frequency of myeloid colonies, with a commensurate decrease in Ery–Meg colonies (Fig. 2c and Extended Data Fig. 3d). This lineage-instructive property of LMO4 was further

supported by induction of ectopic myeloid potential in preMegEs after *Lmo4* overexpression (Fig. 2d). Consistent with the myeloid lineage-instructive capacity of LMO4, both *Lmo4* mRNA (Fig. 2e) and LMO4 protein (Fig. 2f) levels were increased in BEMPs compared with EMPPs.

To confirm this observation in vivo, Kit+ cells isolated from CD45.1 *Gata1*–eGFP mice were transduced with *Lmo4* or control virus, followed by transplantation into lethally irradiated CD45.2 recipients (Fig. 2g). We observed the same level of mononuclear cell engraftment by *Lmo4* and control grafts, whereas *Lmo4* expression significantly decreased the output of platelets and erythrocytes from transduced haematopoietic stem cells (Fig. 2h and Extended Data Fig. 3e,f). Furthermore, analysis of bone marrow cells showed an increased frequency of eosinophils and basophils and decreased erythroid cells (Fig. 2i and Extended Data Fig. 3g,h). Finally, *Lmo4* expression induced a significant increase in the number of BEMPs and decrease in preMegEs (Fig. 2j and Extended Data Fig. 3i), and *Lmo4*-transduced EMPPs and preMegEs were myeloid biased in vitro (Fig. 2k,l and Extended Data Fig. 3j).

To determine whether IL-33 acted directly on EMPPs, we generated bone marrow chimeras between wild-type and IL-33 receptor-knockout cells (IL-33 receptor is also known as ST2 and is encoded by *Il1rl1*) (Fig. 3a). In chimeras, CD45.2 *Il1rl1*-KO cells generated fewer BEMPs than wild-type CD45.2 cells, an effect that was exacerbated by IL-33 administration (Fig. 3b and Extended Data Fig. 4a,b). Microfluidics-based single-cell gene expression profiling of CD45.2 wild-type and *Il1rl1*-KO EMPPs from both control and IL-33-exposed mice using a panel of BEMP-affiliated and Ery–Meg lineage-affiliated genes showed that the EMPP population was coherent and distinct from both BEMPs and preMegEs (Fig. 3c). IL-33 treatment induced a molecular shift of EMPPs towards BEMPs (Fig. 3d), and significantly upregulated *Lmo4* in an ST2-dependent manner (Fig. 3e), with loss of ST2 (and hence IL-33 signalling) leading to the converse shift towards preMegEs (Fig. 3d). ST2 also maintained baseline EMPP *Lmo4* expression (Fig. 3e) and was required for induction of functional myeloid bias in EMPPs (Extended Data Fig. 4c,d). Finally, in vitro IL-33 stimulation of EMPPs showed *Lmo4* upregulation by IL-33 to be NF-κB dependent (Extended Data Fig. 4e). To test whether IL-33 acted similarly in humans, we generated human haematopoietic xenografts in NSGW41 mice and treated these with IL-33 or vehicle, followed by microfluidics-based single-cell expression profiling of lineage-affiliated genes in the CD34+CD38−CD131+ population (Extended Data Fig. 5a), which contains the *GATA1*-expressing progenitors[24]. Clustering (Fig. 3f) and lineage signature analysis (Extended Data Fig. 5b,c) identified the human EMPP (cluster 1, expressing intermediate levels of both BMCP (basophil–mast cell progenitor) and MEP (megakaryocyte–erythroid progenitor) signatures), BEMP and preMegE equivalents, and showed that IL-33 treatment upregulated LMO4 expression in hEMPPs (Fig. 3g) and expanded hBEMPs (Fig. 3h). Accordingly, IL-33-treated human CD131+ common myeloid progenitors (CMPs) generated more myeloid-restricted colonies (Fig. 3i and Extended Data Fig. 5d–f). Therefore, the IL-33–LMO4 axis is an evolutionarily conserved inducer of type 2 myelopoiesis.

## *Lmo4* induces GATA factor reallocation

To address how *Lmo4* promotes myeloid fate, we performed RNA-seq and assay for transposase-accessible chromatin using sequencing (ATAC-seq) of *Lmo4*-expressing and control EMPPs isolated from mice transplanted with lentivirally transduced Kit+ progenitors. GSEA analysis showed that EMPPs expressing *Lmo4* were enriched for the BEMP signatures and depleted for the preMegE signatures at both the transcriptional and the chromatin level (Extended Data Fig. 6a,b and Supplementary Table 5), indicating coordinated transcriptional and epigenetic reprogramming. *Lmo4*-induced changes to transcription factor motif occupancy were inferred from ATAC-seq-based digital footprinting using TOBIAS[25]. This showed that *Lmo4* expression led to a global decrease in

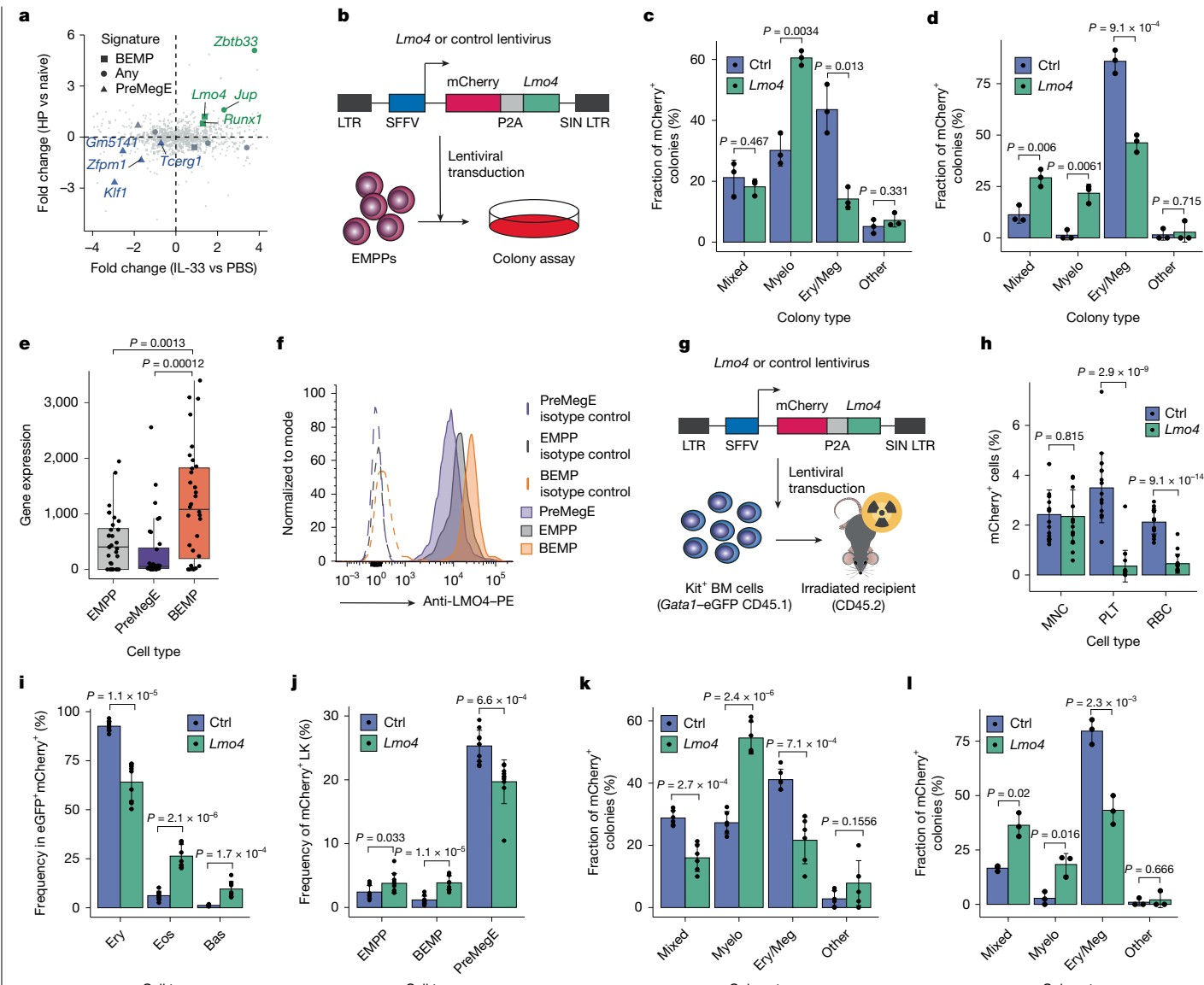

**Fig. 2 | *Lmo4* expression instructs myeloid fate in EMPPs. a**, Differential EMPP transcription factor gene expression between mice treated with IL-33 or PBS (2 days, *n* = 4 per group) and *H. polygyrus* (HP)-infected versus naive mice (4 days, *n* = 3 per group). The shapes indicate significant upregulation (green) or downregulation (blue) in both comparisons (*P* < 0.05). The shapes identify genes from BEMP (square) or preMegE signatures (triangle), or neither (circle). **b**, Experimental workflow for colony assays of EMPPs from *Gata1*–eGFP mice transduced with *Lmo4*-expressing (Lmo4) or control lentivirus. LTR, long-terminal repeat; SFFV, spleen focus forming virus; SIN, self-inactivating. **c**, Frequency of clonal EMPP fates from **b**; three independent experiments. Ctrl, control. **d**, Clonal fates of preMegEs transduced with *Lmo4*-expressing or control lentivirus quantified as in **c**; three independent experiments. **e**, Quantification of *Lmo4* gene expression in single EMPPs, preMegEs and BEMPs using quantitative PCR with reverse transcription (RT–qPCR); *n* = 32 cells per condition. **f**, Histogram showing LMO4 protein levels measured by intracellular flow cytometry in EMPPs, BEMPs and preMegE progenitors. Corresponding isotype controls are shown. The graph is representative of three independent experiments. **g**, Experimental workflow for transplantation of Kit⁺ bone marrow cells transduced with *Lmo4*-expressing or empty backbone (control) lentivirus. **h**, Frequency of mCherry⁺ mononuclear cells (MNCs), red blood cells (RBCs) and platelets (PLTs) in peripheral blood of mice from **g**; *n* = 19 per condition, 3 independent experiments. **i**, Quantification of *Gata1*⁺ blood cell types in mCherry⁺ bone marrow cells from mice in **h**; *n* = 9 from 3 independent experiments. **j**, Frequency of progenitor cell populations within the LK cell fraction in the bone marrow of mice transplanted as in **g**; *n* = 10 per condition from 3 independent experiments. **k**, Frequency of clonal EMPP fates from Extended Data Fig. 3j quantified as in **c**. For *Lmo4*, *n* = 54; for control, *n* = 61 from 6 independent experiments. **l**, Clonal fates of preMegEs quantified as in **k**; three independent experiments. Values are mean ± s.d. *P* values were determined by two-tailed Welch's *t*-test unless otherwise specified.

GATA motif binding in EMPPs, whereas occupancy of RUNX, SPI1/SPIB (that is, PU.1 binding) and JUN (AP-1) motifs was significantly increased (Extended Data Fig. 6c and Supplementary Table 6). Similar analysis of BEMP-specific and preMegE-specific chromatin domains showed increased GATA motif occupancy of BEMP chromatin (Extended Data Fig. 6d and Supplementary Table 7) and decreased occupancy of pre-MegE chromatin (Extended Data Fig. 6e and Supplementary Table 8). Therefore, global GATA motif occupancy and GATA binding to BEMP chromatin were discordantly regulated (Fig. 3j), indicating that LMO4 reallocates GATA factors to BEMP chromatin in EMPPs, with no change to the mRNA levels of *Gata1* and *Gata2* in any of the populations investigated (Extended Data Fig. 6f). To directly test this, we expressed *Lmo4* in the mouse HPC-7 progenitor cell line[26] (Extended Data Fig. 6g), a well-established cell line model for studying transcription factor colocalization in haematopoietic progenitors[27], and measured GATA2 chromatin distribution by chromatin immunoprecipitation followed

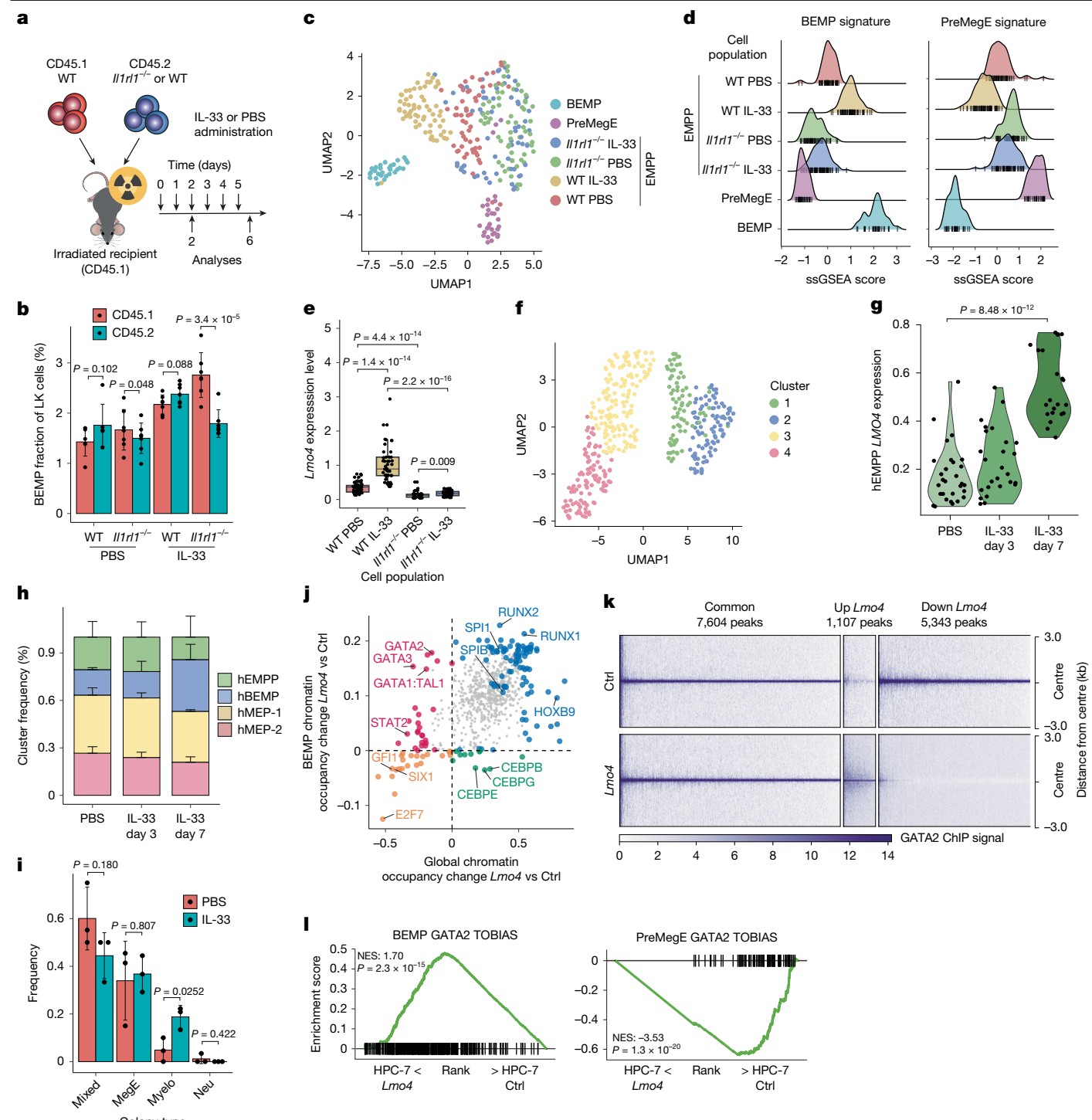

**Fig. 3 | *Lmo4* reallocates GATA2. a**, Generation of bone marrow chimeras with *Il1rl1*^−/−^ or wild-type (WT) CD45.2 and wild-type CD45.1 bone marrow. Reconstituted recipients were IL-33 treated as indicated. **b**, Frequency of BEMPs within CD45.1⁺ and CD45.2⁺ LK populations in chimeras from **a** treated with IL-33 or PBS for 2 days. *n* = 7 per condition except WT PBS (*n* = 6) from 2 experiments. *P* values compare CD45.1⁺ and CD45.2⁺ BEMP frequencies. **c**, UMAP projection of single cells of the indicated cell population using RT–qPCR gene expression data. For BEMP and preMegE, *n* = 30; for WT PBS, *n* = 56; for WT IL-33, *n* = 64; for *Il1rl1*^−/−^ PBS, *n* = 62; and for *Il1rl1*^−/−^ IL-33, *n* = 62. **d**, Plot showing ssGSEA scores of BEMP and preMegE signatures in single cells from **c**. **e**, *Lmo4* gene expression in a single EMPP from **c** for the indicated conditions. *P* values were determined using two-tailed Wilcoxon rank-sum test. **f**, UMAP projection of clustered human CD34⁺CD38⁻CD131⁺ cells from xenografted mice (*n* = 3 per condition) injected with PBS or human recombinant IL-33 for 3 or 7 days using RT–qPCR gene expression data. **g**, Quantification of *LMO4* gene expression in

human EMPPs (hEMPPs; cluster 1) from **f** for the indicated conditions. For PBS, *n* = 29; for IL-33 day 3, *n* = 26; and for IL-33 day 7, *n* = 21. *P* values are as in **e**. **h**, Fraction of clusters in **f** for each indicated condition. Clusters were annotated as follows: hEMPP (1), hBEMP (2), hMEP-1 (3) and hMEP-2 (4). Error bars denote s.d. (*n* = 3). **i**, Quantification of clonal fates of human CD131⁺ CMPs cultured with human IL-33 or PBS; three independent experiments. For PBS, *n* = 71; for IL-33, *n* = 72. **j**, Comparison of differential transcription factor motif occupancy within global chromatin and BEMP-specific chromatin regions using TOBIAS. The large shapes indicate motifs showing significant binding differences. **k**, Heatmap representing the GATA2 ChIP–seq signal in the indicated chromatin regions of control and *Lmo4*-overexpressing HPC-7 cells; three biological replicates. **l**, GSEA comparing GATA2 binding to BEMP or preMegE chromatin accessible in EMPPs, and predicted to be bound by GATA2 using TOBIAS, in data from **k**. NES and *P* values are shown. Values are mean ± s.d. *P* values were determined by two-tailed Welch's *t*-test unless otherwise specified.

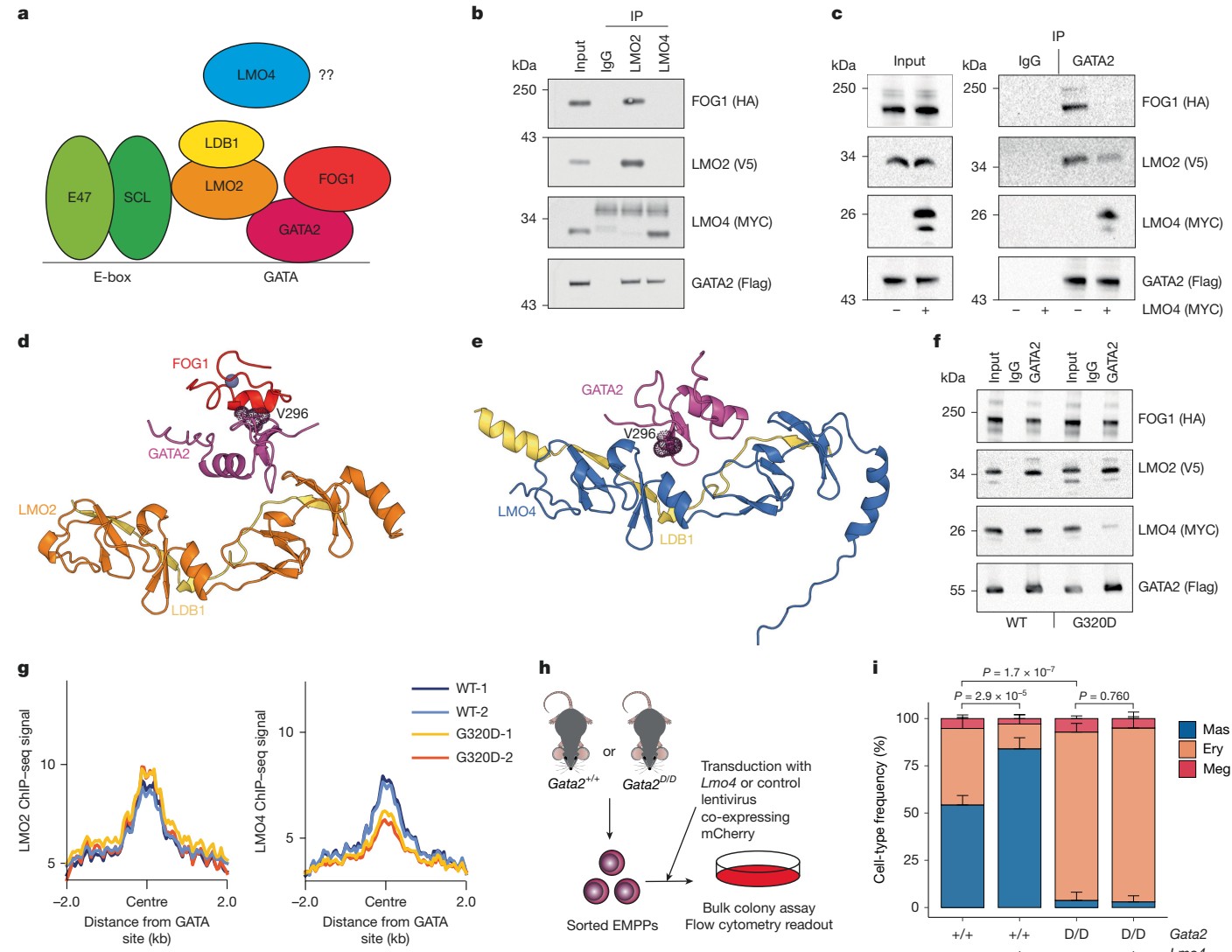

**Fig. 4 | GATA2(G320D) mutation blocks LMO4 interaction. a**, The GATA2 complex containing FOG1, LMO2, LDB1 and SCL–E47, capable of binding to a combined GATA–E-box motif. The effect of LMO4 on this complex is not known. **b**, HA–FOG1, V5–LMO2, MYC–LMO4 and Flag–GATA2 were co-expressed in HEK293T cells, and co-immunoprecipitated with anti-V5, anti-MYC or control IgG, followed by western blotting with the indicated antibodies (Abs). Input denotes 0.75% of IP material. **c**, HA–FOG1, V5–LMO2 and Flag–GATA2 were co-expressed in HEK293T cells in the presence or absence of MYC–LMO4 as indicated, and co-immunoprecipitated with anti-Flag (GATA2), antibody or control IgG, followed by western blotting with the indicated anti-tag antibodies. Input represents 0.75% of the material used for IP. **d**, The structure of LMO2 (amino acids 1–158; orange), LDB1 LID (337–375; yellow) and GATA2 ZnF1 (291–339; magenta) as modelled by AlphaFold3. GATA2 V296 is highlighted with a mesh surface. FOG1 ZnF1 (red) was subsequently docked onto the structure. **e**, The

structure and binding interface of LMO4 (1–165; blue), LDB1 LID (337–375; yellow) and GATA2 N-terminal zinc finger (magenta) as modelled by AlphaFold3. GATA2 V296 is highlighted with a mesh surface. **f**, HA–FOG1, V5–LMO2 and MYC–LMO4 were co-expressed in HEK293T cells with either wild-type or G320D-mutant Flag–GATA2 as indicated, and co-immunoprecipitated with anti-Flag (GATA2) antibody or control IgG, followed by western blotting with the indicated anti-tag antibodies. Input denotes 0.75% of IP material. **g**, Plot of the LMO2 (left) or LMO4 (right) ChIP–seq signal performed in HEK293T cells co-transfected as in **f**; two biological replicates per condition. **h**, Workflow for the colony-forming assay of lentivirally transduced EMPPs. **i**, Mas, Ery and Meg frequency in mCherry⁺ cells from pooled colonies of EMPPs transduced as shown in **h**; five biological replicates per condition. Values are mean ± s.d. $P$ values (determined by two-tailed Welch's $t$-test) are shown for Mas prevalence. Western blots are representative of three experiments.

by sequencing (ChIP–seq). Consistent with in silico footprinting, this showed large-scale GATA2 redistribution upon *Lmo4* expression (Fig. 3k), with GATA2 being lost from preMegE chromatin and enriched within BEMP chromatin (Fig. 3l). In particular, *Lmo4* expression increased GATA2 binding to the *Lmo4* gene itself, and depleted GATA2 from the *Zfpm1* gene (which encodes FOG1; Extended Data Fig. 6h).

## LMO4 displaces FOG1 from GATA2

As LMO4 is known to interact with GATA factors, these results were consistent with LMO4 altering the chromatin distribution of GATA

factors through protein–protein interaction. Although both *Gata1* and *Gata2* are expressed in the Ery–Meg and BEM lineages, we previously found that *Gata1* is dispensable for BEMP commitment[8], whereas GATA2 has been shown to instruct BEM lineage differentiation[12].

GATA factors are found in complexes that include SCL, E47, FOG1 and LMO2 (ref. 13), with LMO2 and FOG1 capable of simultaneously interacting with GATA1 ZnF1 (ref. 28) (Fig. 4a). To address how LMO4 affects the GATA2 ZnF1 interactome, we co-expressed GATA2, FOG1, LMO2 and LMO4 in HEK293T cells using different epitope tags for each protein, and performed co-immunoprecipitation (co-IP) using the LMO2 and LMO4 tags. Anti-LMO2 IP pulled down both FOG1 and GATA2,

confirming the presence of a GATA2–LMO2–FOG1 complex. However, no LMO4 was detected in anti-LMO2 co-IPs. Conversely, neither FOG1 nor LMO2 were detected in anti-LMO4 co-IPs (Fig. 4b and Extended Data Fig. 7a), indicating that LMO4 binding to GATA2 is mutually exclusive with LMO2 and FOG1. To further explore this, we performed anti-GATA2 IP of GATA2–LMO2–FOG1 in the presence or absence of LMO4, finding that LMO4 was able to displace both LMO2 and FOG1 from the GATA2 ZnF1 (Fig. 4c and Extended Data Fig. 7b). Modelling the GATA2–LMO2–LDB1 complex using AlphaFold3 (ref. 29) showed that consistent with NMR-based structure prediction[28], GATA2 V296, critical for FOG1 interaction, is exposed and accessible for binding (Fig. 4d). By contrast, in the predicted GATA2–LMO4–LDB1 complex, V296 is internal and inaccessible to FOG1 (Fig. 4e). The LMO2–GATA2 contact occurred through the LMO2 LIM-2 domain. Replacing the LMO2 LIM-2 domain with the corresponding LMO4 sequence (Extended Data Fig. 7c) was sufficient to make V296 inaccessible (Extended Data Fig. 7d), confer myeloid lineage-instructive capacity on LMO2 (Extended Data Fig. 7e) and block FOG1 binding to the LMO2–GATA2 complex (Extended Data Fig. 7f,g), further linking these activities.

The observation that LMO2 and LMO4 form structurally distinct complexes with GATA2 raised the possibility that the LMO4–GATA2 interaction could be selectively disrupted. We previously generated and characterized a mouse line carrying a GATA2 ZnF1 point mutation (G320 to D; Gata2*D* allele)[30] that is commonly observed in patients with acute erythroleukaemia[31], and found that in myeloid progenitors, this mutation increased accessibility of GATA motifs, while decreasing the accessibility of SPI1/SPIB motifs[30], the converse effect of that observed after ectopic *Lmo4* expression. Co-IP of GATA2 with LMO2, LMO4 and FOG1 showed that the G320D mutation selectively disrupted the LMO4–GATA2 interaction, with both LMO2–GATA2 and FOG1–GATA2 interactions unaffected (Fig. 4f and Extended Data Fig. 8a), and ChIP–seq of GATA2, LMO2 and LMO4 in HEK293T cells transfected with either wild-type or mutant GATA2 showed a selective loss of LMO4 association with GATA2(G320D)-bound sites (Fig. 4g). Furthermore, in vitro differentiation of EMPPs isolated from *Gata2*+/+ or *Gata2*G320D/G320D (*Gata2*D/D) mice, with or without *Lmo4* transduction (Fig. 4h), showed that *Gata2*D/D EMPPs were highly deficient in myeloid colony formation and completely refractory to *Lmo4* fate instruction (Fig. 4i). Finally, knockdown of *Lmo4* by viral transduction in Kit+ bone marrow cells followed by transplantation and IL-33 treatment (Extended Data Fig. 8b,c) impaired BEMP formation (Extended Data Fig. 8d) and EMPP myeloid colony formation (Extended Data Fig. 8e). Together, these results demonstrate a critical role of the LMO4–GATA2 interaction in both baseline and emergency type 2 myelopoiesis.

## LMO4 interaction reallocates GATA2

Characterization of the bone marrow (Fig. 5a and Extended Data Fig. 9a,b) of *Gata2*D/D mice showed a significant decrease in the number of eosinophils and basophils compared with *Gata2*+/+ and *Gata2*D/+ controls. Similarly, we observed a decreased frequency of BEMPs and increased frequency of preMegEs in *Gata2*D/D compared with *Gata2*+/+ mice (Fig. 5b). To determine whether the observed phenotype was intrinsic to the haematopoietic system, we transplanted total bone marrow cells from *Gata2*+/+ and *Gata2*D/D mice into lethally irradiated recipients and analysed their bone marrow (Fig. 5c). This analysis showed a decrease in the number of eosinophils and basophils in the donor cell fraction derived from *Gata2*D/D compared with *Gata2*+/+ mice (Fig. 5d), associated with a decreased frequency of BEMPs and increase of preMegEs (Fig. 5e). A similar phenotype was observed in the spleen (Extended Data Fig. 9c,d).

Comparison of *Gata2*+/+ and *Gata2*D/D EMPPs by GSEA using RNA-seq and ATAC-seq data showed that loss of the GATA2–LMO4 interaction led to decreased expression of both transcriptional and epigenetic BEMP signatures, with a corresponding increase in preMegE signature expression (Extended Data Fig. 9e,f and Supplementary Table 9). Consistent with this, *Gata2*D/D EMPPs acquired transcriptional Ery–Meg bias, while remaining distinct from preMegEs (Fig. 5f,g), and digital transcription factor footprinting showed a global increase in GATA motif occupancy in *Gata2*D/D, compared with *Gata2*+/+, EMPPs (Extended Data Fig. 9g and Supplementary Table 10). Furthermore, comparison of global and BEMP-specific chromatin showed a selective loss of GATA motif occupancy within the BEMP chromatin domain in the absence of GATA2–LMO4 interaction (Fig. 5h, Extended Data Fig. 9h and Supplementary Table 11), whereas preMegE-specific chromatin showed a significant increase in GATA motif occupancy (Extended Data Fig. 9i and Supplementary Table 12). Finally, after infection of *Gata2*D/D mice and wild-type littermates with *H. polygyrus*, we observed a significant increase in the number of worms and worm settlement sites in the small intestine of mutant mice (Fig. 5i,j), showing that GATA2–LMO4 interaction is required for parasite control.

Collectively, these data confirmed that LMO4 binding to GATA2 is critical for the lineage-instructive properties of LMO4: in the absence of the GATA2–LMO4 interaction, GATA factor recruitment to BEMP-specific chromatin is selectively decreased in EMPPs and commitment to a BEMP fate, subsequent myeloid differentiation and anti-parasite immunity severely impaired.

## Discussion

The presence of a dedicated cellular pathway of type 2 myelopoiesis raised the possibility of coordinated intrinsic and extrinsic regulation of BEM production. Here we identified LMO4 as a GATA2 co-factor that is upregulated during BEMP commitment, which has the capacity to promote the generation of all BEM cell types. Furthermore, *Lmo4* expression is upregulated in the upstream EMPPs after *H. polygyrus* infection or direct IL-33 administration, and *Lmo4* expression is sufficient to bias EMPPs towards BEMP commitment. IL-33 also upregulated *Lmo4* expression in human EMPPs, inducing myeloid fate bias. LMO4 is therefore a conserved master regulator of type 2 myelopoiesis that initiates BEM production at the oligopotent progenitor level in response to type 2 inflammatory signalling, which may, in addition to the observed direct IL-33 signalling to EMPPs, involve indirect effects of IL-33-induced cytokines[32]. Emergency production of erythrocytes, platelets, monocytes and neutrophils involves induction of bias in multipotent and oligopotent haematopoietic stem and progenitor cells by cytokines in response to inflammatory or infectious challenge[33,34]. However, in experimental models of basophilia[6] and eosinophilia[7], expansion of committed progenitor populations has been observed, but regulation of BEM fate commitment has not been reported, probably because the relevant progenitor populations were not well defined. EMPPs and BEMPs constitute a relatively small fraction of the commonly used CMP and GMP progenitor definitions[8,35], which could obscure changes to their abundance if these populations were used as reference. It will therefore be important to re-evaluate the action of other relevant cytokines such as KitL, IL-3, IL-5 and GM-CSF, all of which promote the differentiation of one or more BEM cell types[36,37].

The haematopoietic system has long served as a paradigm for cell-fate regulation, and GATA2 is a master regulator of haematopoiesis[38]. Consequently, *Gata2* ablation[39], dysregulation[39] or pathogenic GATA2 mutations[8,40] all have profound and pleiotropic effects on gene regulation and haematopoietic lineage development. Understanding how dynamic changes to GATA2 protein–protein interactions execute cellular decisions and establish cell-type-specific gene expression patterns is therefore central to understanding both normal and defective blood cell production. Here we found that regulation of the GATA2 ZnF1 interactome controls EMPP fate. Both biochemical analysis and structural modelling showed that the binding of LMO4 (but not LMO2) to GATA2 ZnF1 was incompatible with the GATA2–FOG1 interaction.

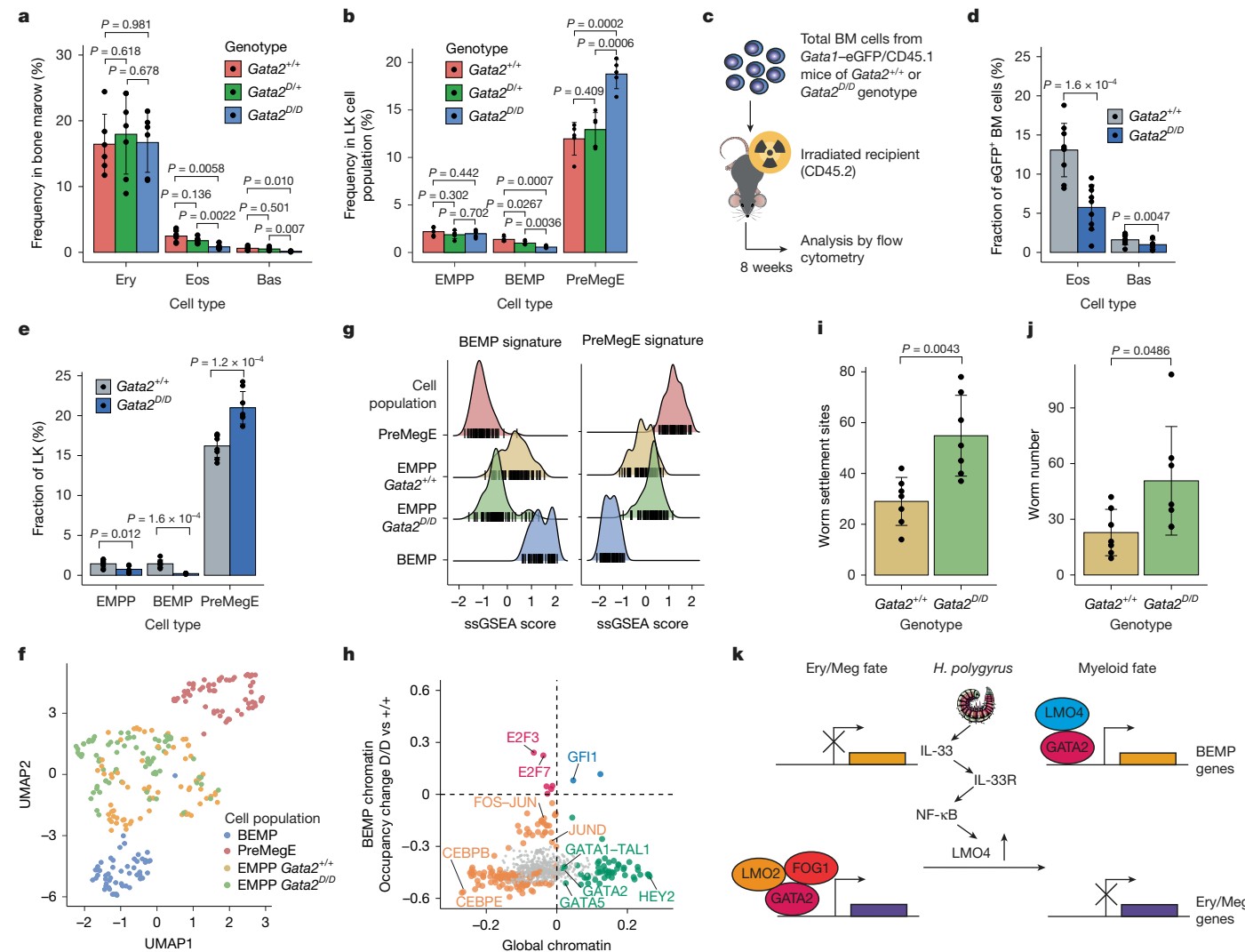

**Fig. 5 | GATA2–LMO4 promotes parasite control. a**, Frequency of Ery, Eos and Bas cells in the bone marrow of *Gata1*–eGFP mice with a *Gata2^{+/+}* (*n* = 6), *Gata2^{D/+}* (*n* = 7) or *Gata2^{D/D}* (*n* = 6) genotype; 2 independent experiments. **b**, Frequency of EMPP, BEMP and preMegE cell populations within the LK bone marrow cell fraction in genotypes from **a**. *n* = 5 per genotype; 2 independent experiments. **c**, Workflow for transplantation of bone marrow (BM) from *Gata1*–eGFP/CD45.1 mice with a *Gata2^{+/+}* or *Gata2^{D/D}* genotype into lethally irradiated CD45.2 recipient mice. **d**, Frequency of Eos and Bas in the eGFP^+ bone marrow cell fraction of mice transplanted as in **c**. *n* = 9 per genotype; 2 independent experiments. **e**, Frequency of EMPPs, BEMPs and preMegEs within the LK bone marrow cell fraction of mice transplanted as in **c**. *n* = 8 per genotype; 2 independent experiments. **f**, UMAP projection of single cells of the indicated cell population, generated using microfluidics RT–qPCR gene expression data. For BEMPs, *n* = 55; for preMegEs, *n* = 63; for *Gata2^{+/+}* EMPPs, *n* = 71; and for

*Gata2^{D/D}* EMPPs, *n* = 75; 2 independent experiments. **g**, Plot showing the ssGSEA enrichment scores of BEMP and preMegE signatures in single cells from **f**. **h**, Comparison of differential transcription factor motif occupancy within global chromatin and BEMP-specific chromatin regions using TOBIAS. The large shapes indicate motifs showing significant binding differences. **i**, Quantification of worm settlement sites in the intestine of *Gata2^{+/+}* and *Gata2^{D/D}* mice infected with *H. polygyrus* for 14 days. *n* = 7; 3 experiments. **j**, Quantification of worm number in the intestine of mice from **i**. **k**, Model for induction of type 2 myelopoiesis by helminth infection. IL-33 released from cells damaged by infection induces *Lmo4* in oligopotent EMPPs via NF-κB, leading to displacement of FOG1 and LMO2 from GATA2, GATA2 reallocation from Ery–Meg to BEM chromatin, and myeloid fate. Values are mean ± s.d. *P* values were determined by two-tailed Welch's *t*-test unless otherwise specified.

Furthermore, cells containing GATA2(G320D), which is defective in LMO4 binding, were refractory to LMO4-mediated myeloid reprogramming and defective in BEM generation. Therefore, LMO4 modulates the GATA2 proteome to induce EMPP lineage bias and myeloid differentiation. Given the essential role of FOG1 in Ery–Meg lineage commitment[16], FOG1 displacement from GATA2 provides a simple mechanism by which LMO4 inhibits EMPP commitment to Ery–Meg fate, and conversely, the ability of FOG1 to inhibit both eosinophil[17] and mast cell[18] differentiation can now be explained by competition between LMO4 and FOG1 for GATA2 ZnF1 interaction. Overall, this identifies the LMO4–GATA2 interaction as a specific and essential requirement for type 2 myelopoiesis, and interference with LMO4 or

its interaction with GATA2 as a molecular strategy for selective modulation of type 2 inflammation.

A key consequence of LMO4–GATA2 interaction was reallocation of GATA factors, and GATA2 in particular, from preMegE-specific to BEMP-specific chromatin. GATA motif occupancy within BEMP chromatin increased after LMO4 expression, even as global GATA motif occupancy decreased. Conversely, in GATA2(G320D)-mutant EMPPs, GATA motif occupancy increased both globally and within preMegE chromatin, whereas decreased within BEMP chromatin, demonstrating that GATA factor allocation is an active mechanism, a conclusion further supported by the significant GATA2 reallocation observed in HPC-7 cells upon LMO4 expression. Reallocation of Runx1 by PU.1 has

previously been shown to coordinate gene expression during thymocyte differentiation[41], and to control the balance between differentiation and stemness in myeloid progenitors[42]. This type of mechanism can therefore control multiple processes during blood formation. Furthermore, BEMP-specific transcription in EMPPs was correlated to GATA factor allocation, indicating that GATA factor chromatin distribution directly determines lineage commitment and is controlled by competition between lineage-specific GATA co-factors, a notion supported by the observation that FOG1 is capable of displacing GATA1 from mast cell-specific regulatory elements when ectopically expressed in mast cells[43]. Given the ability of LDB1-containing transcription factor complexes to mediate long-range interactions[44], this is likely to involve both changes to local transcription factor occupancy and reorganization of the 3D chromatin structure.

In addition to blood, GATA factors are central to the development of endothelium, skin, neurons, heart and multiple epithelia, functions executed through their participation in diverse, tissue-specific protein complexes[13]. The principles herein identified are therefore likely to apply to other GATA co-factor-regulated developmental processes, including neuronal specification[21]. Furthermore, our finding that GATA distribution is regulated via the *H. polygyrus*–IL-33–*Lmo4* axis (Fig. 5k) indicates that GATA-regulated fate decisions can be directly linked to the physiological needs of the organism, identifying GATA factor chromatin reallocation as a novel stress response mechanism.

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

# Methods

## Animal studies

All animal studies were performed in accordance with UK Home Office regulations with approval by the University of Oxford Animal Welfare and Ethical Review Body (project license number 30/3359, PP2240412 and PP3246892) and the Francis Crick Institute Animal Welfare and Ethical Review Body (project license number PP8468807), as well as the Irish Health Products Regulatory Authority regulations License AE19136/P108 and P196, approved by Trinity College Dublin's Animal Research Ethics Committee. NOD.Cg-Kit[W-41J]Tyr[+]Prkdc[scid]Il2rg[tm1Wjl]/ThomJ (NSGW41) mice[45], Il1rl1-KO mice[46], Gata2[G320D] knock-in mice[30] and Gata1–eGFP BAC transgenic mice[8] have been previously described. For IL-33 administration 9-week-old male C57BL/6J or Gata1–eGFP mice or short hairpin RNA (shRNA)-targeting Scramble or Lmo4, or control or St2[−/−] transplanted mice were intraperitoneally injected with 1 μg of recombinant mouse IL-33 (Peprotech) or PBS. For parasite infection, 9–12-week-old male C57BL/6J were used. Eight-nine-week-old Gata2[D/D] mice and wild-type littermates were infected with 200 H. polygyrus third-stage larvae, obtained from faecal cultures of H. polygyrus-infected mice, by oral gavage. Mice were maintained under specific pathogen-free conditions at 19–23 °C and 45–65% humidity on a 12-h light–12-h dark cycle. Sex and age-matched animals were randomly selected and allocated to experimental group, expect when assignment based on mouse genotype. Experiment and data collection were not performed blind to the condition of the experiment.

## Plasmid construction

Flag–LMO2, MYC–LMO4, shRNA-targeting Scramble and shRNA-targeting Lmo4 constructs were obtained from Addgene (#64893, #22965, #59299 and #59292, respectively). The HA–FOG1 construct has been previously described[17]. Gata2 as well as LMO2/4 chimeras LMO-C1 and LMO-C2 cDNA were synthetized and subcloned into pLeGO-G (Addgene #27347) or pLeGO-C2 (Addgene #27339) under a mCherry-P2A reporter. For lentiviral transduction, Lmo4 cDNA was amplified and subcloned into pLeGO-C2 (Addgene #27339) or LT3-GEPIR (Addgene #111177) under a mCherry-P2A reporter.

## Lentiviral production

HEK-293T (American Type Culture Collection CRL-11268) cells were plated in DMEM (Gibco) supplemented with 10% FBS (Gibco), 2 mM L-glutamine (Gibco) and 100 U ml[−1] penicillin–streptomycin (Gibco), a day before co-transfection. Co-transfection was performed using PEIpro (Polyplus) according to the manufacturer's recommendation, with psPAX2 (Addgene #12260), pMD2-G (Addgene #12259) and LeGO-C2-P2A-Lmo4/empty vectors. The viral supernatant was collected 48 h post-transfection, filtered using 0.45-μm cellulose acetate filter (Milipore) and concentrated using an Optima XPN-80 (Beckman Coulter) at 23.000 rpm for 1.5 h at 4 °C. HEK-293T cells were purchased from the supplier (Sigma) as a cell line authenticated by STR profiling, and was tested negative for mycoplasma.

## Bone marrow transplantation

A total of $5 \times 10^5$ bone marrow cells were isolated from 12-week-old female Gata1–eGFP and Gata2[+/+] or Gata2[D/D] mice and engrafted in lethally irradiated (10 Gy) 8–10-week-old female CD45.2 recipient mice by intravenous injection. To generate Il1rl1[−/−] chimeras, $5 \times 10^5$ total bone marrow cells from CD45.2 Il1rl1[−/−] or wild-type control mice were co-transplanted with an equal number of wild-type CD45.1 bone marrow cells into lethally irradiated (10 Gy) 8–10-week-old female CD45.1 recipient mice by intravenous injection.

## Transplantation of genetically modified HSPCs

Bone marrow was isolated from 10–14-week-old female C57BL/6J or Gata1–eGFP mice[8]. Haematopoietic stem and progenitor cells (HSPCs)

were enriched using CD117 MicroBeads and LS columns, according to the manufacturer's recommendation (Miltenyi Biotec). Transduction of empty/Lmo4-expressing or shRNA-targeting Scramble/Lmo4 lentivirus was at a multiplicity of infection of 50 in StemSpan SFEM (STEMCELL Technologies) supplemented with 2 mM L-glutamine (Gibco), 100 U ml[−1] penicillin–streptomycin (Gibco), 5 μg ml[−1] polybrene (Merck), 100 ng ml[−1] mouse SCF (Peprotech) and 100 ng ml[−1] human TPO (Peprotech). Cells were spinfected at 33 °C and 700g for 1 h and incubated at 37 °C for 7 h or overnight. A total of $5 \times 10^5$ cells were injected intravenously into lethally irradiated (12 Gy; split dose) 8–10-week-old female CD45.2-recipient or CD45.1-recipient mice.

## Colony-forming unit assay

Sorted cells (100–5,000) were plated in MethoCult GF M3434 (STEMCELL Technologies) supplemented with 25 ng ml[−1] human TPO (Peprotech), 50 ng ml[−1] mouse IL-5 (Peprotech), 50 ng ml[−1] mouse IL-9 (Peprotech) and 100 U ml[−1] penicillin–streptomycin (Gibco). Bulk or single-colony analysis was performed 10 days after cell plating. For single-colony readout, individual colonies were picked up in PBS (Gibco) supplemented with 5% FBS (Gibco) in 96-well plates and analysed by flow cytometry.

## Human single-cell culture and xenotransplantation

Bone marrow from healthy volunteer was purchased from Lonza (Lonza Bioscience). CMP CD131[+] was isolated by flow cytometry and cultured at the single-cell level in round-bottom 96-well plates in 50 μl of StemSpan (STEMCELL Technologies) with 20 ng ml[−1] human SCF, 20 ng ml[−1] human Flt3L, 20 ng ml[−1] human IL-3, 50 ng ml[−1] human IL-5, 20 ng ml[−1] human IL-6, 50 ng ml[−1] human GM-CSF, 20 ng ml[−1] human G-CSF, 40 μg ml[−1] human LDL (Sigma-Aldrich), 0.5 U ml[−1] erythropoietin, 100 ng ml[−1] thrombopoietin with 100 U ml[−1] penicillin–streptomycin (Gibco), 0.1 mg ml[−1] streptomycin and with or without 100 ng ml[−1] human IL-33. Cytokines supplied by Peprotech. Lineage analysis was performed by flow cytometry, 20 days after cell plating. For xenotransplantation, $10^4$ CD34[+] cells isolated by flow cytometry were injected in NSGW41 mice by intrafemoral injection.

## ELISA assay

Bone marrow supernatant was collected from tibias and femur by flushing. IL-33 was quantified with mouse/rat IL-33 Quantikine ELISA kit (R&D) according to the manufacturer's recommendation, using CLARIOstar Plus (BMG labtech).

## Flow cytometry analysis and cell sorting

Mouse bone marrow cells were isolated by crushing leg bones in FACS media that consisted of PBS (Gibco) supplemented with 2% FBS (Gibco). For HSPC hierarchy analysis, bone marrow cells were subjected to CD117 enrichment using CD117 MicroBeads and LS column, following the manufacturer's recommendation (Miltenyi Biotec). Spleen cells were obtained by crushing the mouse spleen on a 0.70-μM filter using FACS media. Isolation of peritoneal cavity cells was performed by intraperitoneal injection of PBS (Gibco) in culled mice. After peritoneum massage, peritoneal suspension containing cells was collected, spun down at 500g for 5 min at 4 °C and cell pellet resuspended in FACS media. Peripheral blood was collected from the mouse tail vein into lithium heparin-coated microvettes (Sarstedt). An aliquot was collected for analysis of red blood cells and platelets. For mononuclear cell fraction analysis, samples were subjected to red blood cell lysis using ammonium chloride solution (STEMCELL Technologies) for 10 min in ice. After centrifugation at 500g for 5 min at 4 °C, the cell pellet was resuspended in FACS media. For xenografted human cell sorting, bone marrow cells isolated from NSGW41-transplanted mice were subjected to mCD45/Ter119 depletion using mCD45 and Ter119 MicroBeads and LS column, following the manufacturer's recommendation (Miltenyi Biotec). Cells were incubated in FACS media containing Fc-block antibody

(eBioscience) and subsequently stained with antibody cocktail for 20 min in ice. Samples were washed with FACS media, spun down at 500g for 5 min at 4 °C and resuspended in FACS media supplemented with living/dead cell dye: 1 µg ml⁻¹ 7AAD (Insight Biotechnology) or 0.1 µg ml⁻¹ DAPI (Miltenyi Biotec). Intracellular LMO4 staining was conducted using the True-Nuclear Transcription Factor Buffer Set (BioLegend) according to the manufacturer's protocol. Stained cells were analysed on LSR Fortessa or LSR X-20 flow cytometers (BD Biosciences). Cell sorting was carried out on FACSAria III, FACSAria Fusion or FACSymphony S6 (BD Biosciences) using FACS DIVA software (v9.0). Analysis was performed using FlowJo software (v10.8.1). The antibodies used are listed in Supplementary Table 13.

## Co-IP and western blot

Transfection of HEK293T cells was performed using PEIpro (Polyplus) according to the manufacturer's recommendation. Forty-eight hours post-transfection, cells were washed in PBS (Gibco) and isolated using trypsin–EDTA 0.05% (Gibco). Total cell lysis was performed using RIPA buffer (25 mM Tris-HCl, 150 mM NaCl, 1% NP-40, 1% sodium deoxycholate and 0.1% SDS) supplemented with cOmplete protease inhibitor cocktail (Roche) for 20 min on ice. Samples were sonicated for one cycle (30 s ON/OFF) using Bioruptor Pico (Diagenode) and spun down at 10,000g for 25 min at 4 °C. Proteins in supernatant were quantified using Qubit Protein broad range assay (Invitrogen). The following antibodies were used for immunoprecipitation: anti-GATA2 (R&D), anti-Flag (Sigma and Cell Signaling) or anti-MYC (Invitrogen and Cell Signaling). Immune complexes were purified using protein G magnetic beads (NEB) followed by serial washes in RIPA buffer supplemented with cOmplete protease inhibitor cocktail (Roche). Magnetic beads were resuspended in 2X Laemmli buffer (Bio-Rad) supplemented with 100 mM dithiothreitol (Merck) and incubated 15 min at 96 °C. Beads were separated on a magnetic rack and supernatant collected for western blotting, according to Pierce protocols.

## In vitro signalling inhibitor treatment

Sorted EMPP isolated from Gata1–eGFP mice were plated in round-bottom 96-well plates in 50 µl of StemSpan SFEM (STEMCELL Technologies) supplemented with 100 U ml⁻¹ penicillin and 0.1 mg ml⁻¹ streptomycin, 2 mM L-glutamine, 20 ng ml⁻¹ mouse SCF (PeproTech), 20 ng ml⁻¹ human Flt3-ligand (PeproTech), 20 ng ml⁻¹ human thrombopoietin (PeproTech) and PBS or recombinant IL-33 (PeproTech). Treatment was performed with SP600125 (1 µM; Medchem Express), U0125 (5 µM, Medchem Express), IKK-16 (0.5 µM, Medchem Express) or PBS for 16 h. Equal living cell numbers between treatment groups were sorted by flow cytometry directly in lysis buffer using CellsDirect One-Step RT–qPCR kit, immediately followed by cDNA synthesis and RT–qPCR using TaqMan Gene Expression Master Mix with QuantStudio 3 (Applied Biosystems), according to the manufacturer's protocol.

## Microfluidics single-cell gene expression analysis

Single cells were sorted in a 96-well PCR plate in 5 µl of CellsDirect One-Step RT–qPCR lysis buffer (Life Technologies) supplemented with 0.2X TaqMan probes for each targeted gene (Supplementary Table 14). cDNA synthesis and pre-amplification were conducted with the following program: 15 min at 50 °C, 2 min at 95 °C and 22 cycles of (15 s at 95 °C; 4 min at 60 °C). Samples were diluted in 5 volumes of TE buffer (Invitrogen) and gene expression quantified using the BioMark 192.24 Dynamic Array platform (Fluidigm) according to the manufacturer's protocol.

Gene expression was measured according to ΔCt method using Hprt1 and B2m (for mouse) or HPRT and B2M (for human) as housekeeping genes with the geometric mean normalization method (Supplementary Tables 15–17). Low-quality cells marked by non-detectable housekeeping gene expression were excluded, as well as non-targeted cells characterized by non-detectable expression of Gata1 and

expression of lymphoid genes (Dntt, Irf8, Pax5, Ebf1 or Tcf7). Principal component analysis, UMAP projection and clustering were performed using Seurat (v5)[47]. Single-cell GSEA was performed using ssGSEA (https://rpubs.com/pranali018/SSGSEA).

## RNA-seq

Bulk RNA-seq was performed as previously described[48]. In brief, 100 cells were sorted in lysis buffer containing 0.18% Triton X-100 (Sigma-Aldrich), 4 U RNase inhibitor (Takara), 2.5 mM dNTP (Invitrogen) and 2.5 µM oligo-dT (Biomers.net). Reverse transcription was performed using 50 U SMARTScribe reserve transcriptase (Takara), and cDNA amplification was performed using 50 U SeqAmp Polymerase (Takara). cDNA was purified using SPRIselect beads (Beckman Coulter) and the library prepared using Nextera XT DNA Library Preparation Kit and Nextera XT Index kit (Illumina) according to the manufacturer's recommendation. Libraries were sequenced on Illumina NovaSeq6000 or NovaSeq X Plus (150 bp; paired-end read).

## ATAC-seq

Bulk ATAC-seq was performed based on the Omni-ATAC protocol[49]. In brief, 500–1,000 cells were sorted into tagmentation mix containing 1X TD buffer (Illumina), 0.01% digitonin (Promega) and 0.1% Tween-20 (Bio-Rad), and tagmentation was performed using TDE1 enzyme (Illumina) at 37 °C for 30 min. DNA was purified using the MinElute PCR Purification Kit (Qiagen) and amplified using NEBNext High-Fidelity 2×PCR master mix (NEB) and customized Nextera primers (IDT). PCR product was purified using SPRIselect beads (Beckman Coulter), quantified using the NEBNext Library Quant Kit (NEB) and sequenced on Illumina NovaSeq X Plus (150 bp; paired-end read).

## ChIP–seq

ChIP was performed as previously described[50]. In brief, the HPC-7 cell line was transduced using doxycycline-inducible Lmo4/empty-expressing lentivirus and the transduced cell selected using puromycin. Doxycycline (1 µg ml⁻¹) was added in culture media for 48 h and mCherry⁺ cells sorted by flow cytometry. Cells (1 × 10⁶) were incubated with 2 mM disuccinimidyl glutarate and 1% paraformaldehyde, followed by addition of glycine to 125 mM and incubation for 5 min. Cells were pelleted and washed in PBS (Gibco) supplemented with cOmplete protease inhibitor cocktail (Roche). Pelleted cells were resuspended in lysis buffer A (50 mM HEPES pH 7.5, 140 mM NaCl, 1 mM EDTA, 10% glycerol, 0.5% NP-40 and 0.25% Triton X-100) supplemented with cOmplete protease inhibitor cocktail (Roche) and incubated in ice for 15 min. Nuclei were spun down and washed in lysis buffer B (200 mM NaCl, 1 mM EDTA, 0.5 mM EGTA and 10 mM Tris-HCl pH 7.5), then pelleted and resuspended in wash buffer (50 mM Tris-HCl pH 7.5 and 50 mM CaCl₂) supplemented with cOmplete protease inhibitor cocktail (Roche). Nuclei were pelleted and resuspended in chromatin digestion mix (1X micrococcal nuclease buffer (NEB), 2,000 U micrococcal nuclease (NEB) and 0.1 mg ml⁻¹ BSA) and incubated at 37 °C for 10 min. The reaction was stopped by adding Stop buffer (0.5% SDS, 5 mM EDTA and 5 mM EGTA final concentration) and tubes transferred in ice. Digested nuclei were sonicated for one cycle 30 s ON/OFF using Bioruptor Pico (Diagenode) and diluted in 4 volumes of ChIP dilution buffer (0.01% SDS, 1.1% Triton X-100, 1.2 mM EDTA, 16.7 mM Tris-HCl pH 8.1 and 167 mM NaCl). Relevant antibodies (Supplementary Table 13) were added and incubated at 4 °C overnight. Protein G magnetic beads (NEB) were added and incubated for 2 h at 4 °C. Bead-bound chromatin was washed twice in low-salt buffer (0.1% SDS, 1% Triton X-100, 2 mM EDTA, 20 mM Tris-HCl pH 8.1 and 150 mM NaCl), twice in high-salt buffer (0.1% SDS, 1% Triton X-100, 2 mM EDTA, 20 mM Tris-HCl pH 8.1 and 500 mM NaCl), twice in LiCl buffer (0.25 M LiCl, 1% NP-40, 1% deoxycholic acid, 1 mM EDTA and 10 mM Tris-HCl pH 8.1) and once in TE buffer (1 mM EDTA and 10 mM Tris-HCl pH 8.1). DNA was reverse crosslinked by adding 1.6 U proteinase K (NEB) and incubated at 55 °C for 3 h, then at 65 °C overnight. Finally,

DNA was purified with a 1.8X ratio SPRI clean-up and quantified using the Qubit dsDNA HS Assay Kit (Thermo Fisher Scientific). Sequencing libraries were prepared using the NEBNext Ultra II DNA Library Prep Kit for Illumina (NEB) according to the manufacturer's recommendation, and sequenced aiming 40 million reads per sample in a NextSeq550 or NovaSeq-X Plus (Illumina). The HPC-7 cell line was authenticated by differentiation into erythroid, megakaryocyte and myeloid lineages, and tested negative for mycoplasma.

### RNA-seq analysis
Quality control and adaptor sequences were removed from fastq file using Trim Galore! (v0.6.5) and subsequent reads were quantified with Salmon (v1.2.0)[51] using the mouse mm10 reference genome (https://www.gencodegenes.org/mouse/release_M10.html). Read count normalization and differential gene expression were performed using DESeq2 with default parameters (v1.42.0)[52]. GSEA was performed using the javaGSEA application (v4.2.3)[53] and FGSEA R package (v1.34.2; http://bioconductor.org/packages/fgsea/).

### ATAC-seq analysis
Quality control and adaptor sequences were removed from Fastq files using Trim Galore! (v0.6.5). Reads were aligned to the mm10 reference genome using Bowtie2 (v2.4.1)[54]. Subsequent SAM files were converted to BAM files, and PCR duplicate reads were filtered out using SAMtools (v1.17)[55]. Peak calling was performed using MACS2 (v2.1.2)[56] with the following parameters: -q 0.05 -nomodel --shift -100 -extsize 200. A normalized Bigwig file was generated using bamCoverage (deepTools v3.5.3)[57] and visualized on Integrative Genomics Viewer (v2.19.1)[58]. Unique reads mapped to these regions were quantified using featuresCounts (v2.0.1)[59]. Differential open chromatin regions were measured using DESeq2 (v1.42.0)[52]. For GSEA, peaks were ranked by statistic from DESeq2 to generate a rank-ordered list that was imported into the javaGSEA application (v4.2.3)[53] and FGSEA R package (v1.34.2; http://bioconductor.org/packages/fgsea/). Transcription factor footprint analysis was performed using TOBIAS (v0.12.1)[25] with the JASPAR2020 vertebrate core database[60]. BAM files from replicates were aggregated into a single BAM file and used for transposition insertion bias correction using the ATACorrection function. The transcription factor footprint score within selected open chromatin regions was computed using corrected files with ScoreBigwig function. Finally, differential transcription factor occupancy was quantified using the BINDetect function.

### ChIP–seq analysis
FASTQ files were processed to BAM files similarly as described in ATAC-seq analysis section, using mm10 (https://www.gencodegenes.org/mouse/release_M10.html) or hg19 (https://www.gencodegenes.org/human/release_19.html) reference genomes for experiment using HPC-7 or HEK293T cell lines, respectively. Peak calling was performed using MACS2 (v2.1.2)[56] with the following parameters: -q 0.05 -nomodel -extsize 200. Read counts mapped on these regions were quantified using featureCounts (v2.0.1)[59] and a normalized Bigwig file was generated using bamCoverage (deepTools v3.5.3)[57]. Differential binding was measured using DESeq2 (v1.42.0)[52] and visualized with deepTools (v3.5.3)[57]. GSEA was performed using FGSEA R package (v1.34.2).

### Protein structural modelling
The structures and binding interfaces of the protein complexes were generated using AlphaFold3 (ref. 29). The interaction between LMO2 (residues 1–158), LDB1 LID (residues 337–375) and the GATA2 N-terminal zinc finger (residues 291–339) was modelled, and the binding interface between the GATA2 N-terminal zinc finger and FOG1 ZnF1 was based on the previously determined NMR structure (Protein Data Bank ID 1Y0J) of the GATA1 N-terminal zinc finger complexed with *Drosophila melanogaster* homologue, ush ZnF1 (dFOG; residues 202–235)[61]. Similarly, the complex containing full-length LMO4 (residues 1–165), LDB1

LID and GATA2 N-terminal zinc finger was also modelled using the same method. A third model was generated for the LMO-C2 construct, which included all residues from LMO2 aside from 112–151 of LMO4, in complex with LDB1 LID and GATA-2 N-terminal zinc finger. All models were visualized using the PyMOL Molecular Graphics System (v2.1.0; http://www.pymol.org/pymol), where proteins were rendered as cartoon diagrams.

### Statistics and reproducibility
Statistical analysis and representation were performed using R software. For normally distributed data, two-tailed $t$-tests were applied with Welch's correction, which did not assume equal variance. Normality was tested using the Shapiro–Wilk test. The statistical tests used have been described for each analysis in the corresponding figure legend. Sample sizes were chosen based on the variance previously observed in similar experiments, to allow detection of a 50% difference at $P = 0.05$ with 80% power. Block designs were used to allow variance estimates to be adjusted if necessary. Data collection and analysis were not performed blind to the conditions of the experiments. The boxplots display the median as the centre line of the box, with the box representing the distribution's 25th (minima) and 75th (maxima) percentiles; the whiskers extend up to 1.5 times the interquartile range (Q3–Q1) from the minima and maxima.

### Reporting summary
Further information on research design is available in the Nature Portfolio Reporting Summary linked to this article.

## Data availability
Raw data generated by Illumina sequencing and processed data files are available at Gene Expression Omnibus as super-series GSE276830. Source data are provided with this paper.

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

**Acknowledgements** We thank M. de Bruijn for the HPC-7 cell line; T. Milne for advice on ChIP–seq; the Weatherall Institute of Molecular Medicine (WIMM) flow cytometry facility for assistance with cell sorting; the Oxford University Biomedical Service for assistance with animal experiments; the WIMM single-cell facility for expert assistance; the WIMM CBRG

for computational support; and staff at the Crick Science Technology Platforms for expert support. This work was supported by MRC Unit grant (MC_UU_12009/7 and MC_UU_00029/9) to C.N.; European Hematology Association Junior Research Grant (RG-202112-01553) to A.F.; and by the Francis Crick Institute, which receives its core funding from Cancer Research UK (FC001128 and FC001159), the UK Medical Research Council (FC001128 and FC001159) and the Wellcome Trust (FC001128 and FC001159). V.P. (212300/Z/18/Z) and F.P. (226579/Z/22/Z) acknowledge additional funding from the Wellcome Trust and P.G.F. is supported by Research Ireland (24/FFP-P/12948). The WIMM FACS Core facility is supported by MRC HIU, MRC MHU (MC_UU_12009), NIHR Oxford BRC and the John Fell Fund (131/030 and 101/517), the EPA fund (CF182 and CF170) and by WIMM Strategic Alliance awards (G0902418 and MC_UU_12025).

**Author contributions** C.N. and A.F. conceived the study. C.N., F.P. and V.P. supervised the experiments. A.F., C.D.G., Y.M., R.D., Z.Z., B.Z. and F.P. performed the experiments. P.G.F. generated the *Il1rl1*-knockout mice. E.J.M. performed the structural analysis. C.N. and A.F. performed the bioinformatics analysis, analysed the data and wrote the manuscript.

**Competing interests** The authors declare no competing interests.

**Additional information**
**Correspondence and requests for materials** should be addressed to Claus Nerlov.

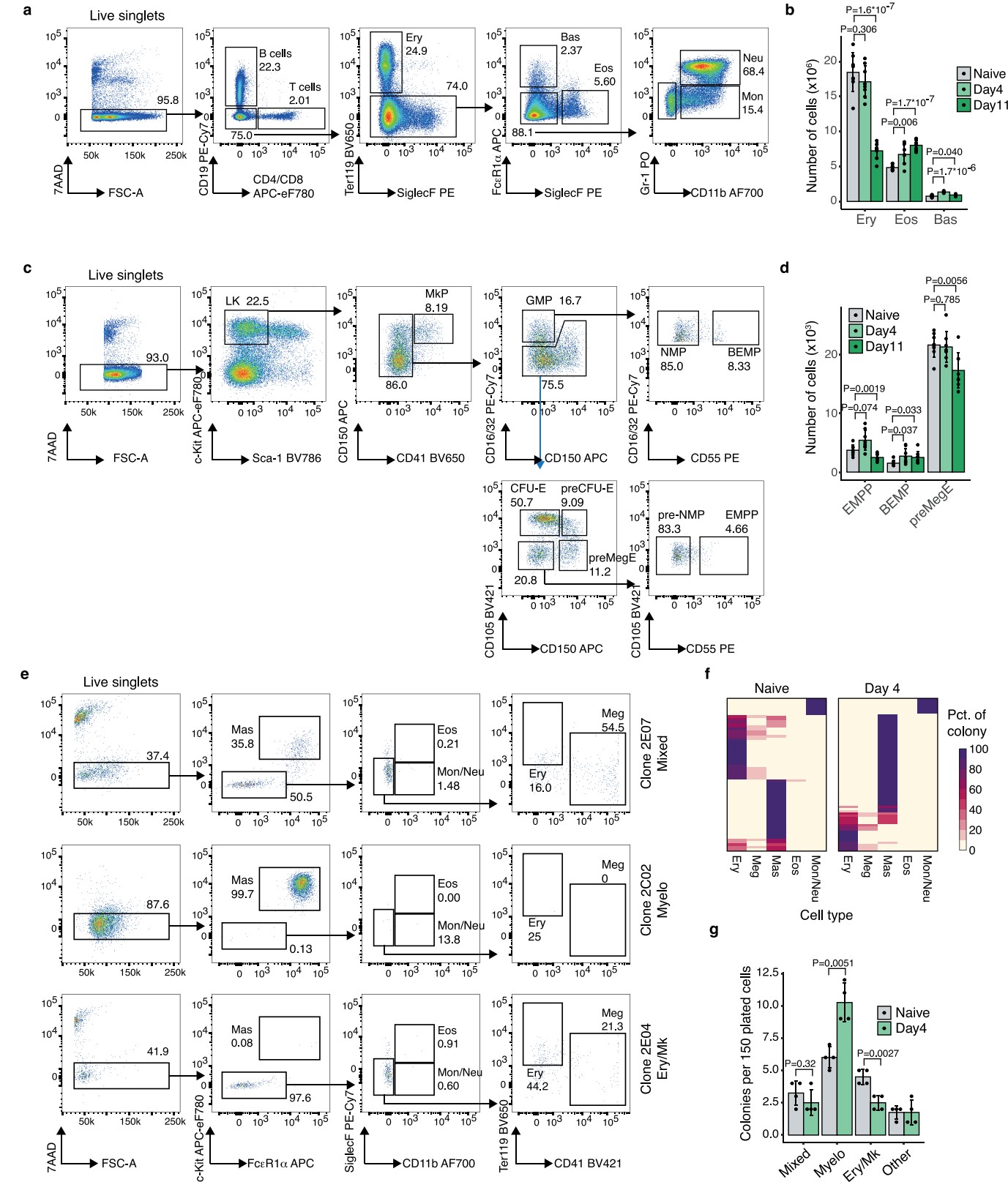

**Extended Data Fig. 1 | Effect of *H. polygyrus* on EMPP lineage potential.**
**a**, Gating strategy used to quantify the frequency of erythroid (Ery),
eosinophil (Eos) and basophil (Bas) cells in mouse BM. **b**, Quantification
of erythroid (Ery), eosinophils (Eos) and basophils (Bas) cell number isolated
from the bone marrow of mice, related to Fig. 1a,b. **c**, Gating strategy used for
sorting EMPPs, preMegEs and BEMPs from mouse BM. **d**, Quantification of
EMPP, BEMP and preMegE cell number isolated from the bone marrow of mice,
related to Fig. 1a,c. **e**, Gating strategy used to analyse single cell colony forming

assay. **f**, Clonal fate heatmap showing the cell type frequency in each colony in
EMPP colonies from day 4 *H.Polygyrus*-infected (n = 68) or naïve mice (n = 62).
6 independent experiments. Ery, Meg, Mas, Eos and Mon/Ne) frequencies
were quantified by flow cytometry. **g**, Quantification of clonal fates from (f) as
colonies per 150 plated cells. Myelo: Mas and/or Eos; Ery/Meg: Ery and/or Meg
colonies; Mixed: Myelo and Ery/Meg. Other: Mon/Neu. Values are mean ± SD.
Statistical significance was determined by two-tailed Welch's t-test.

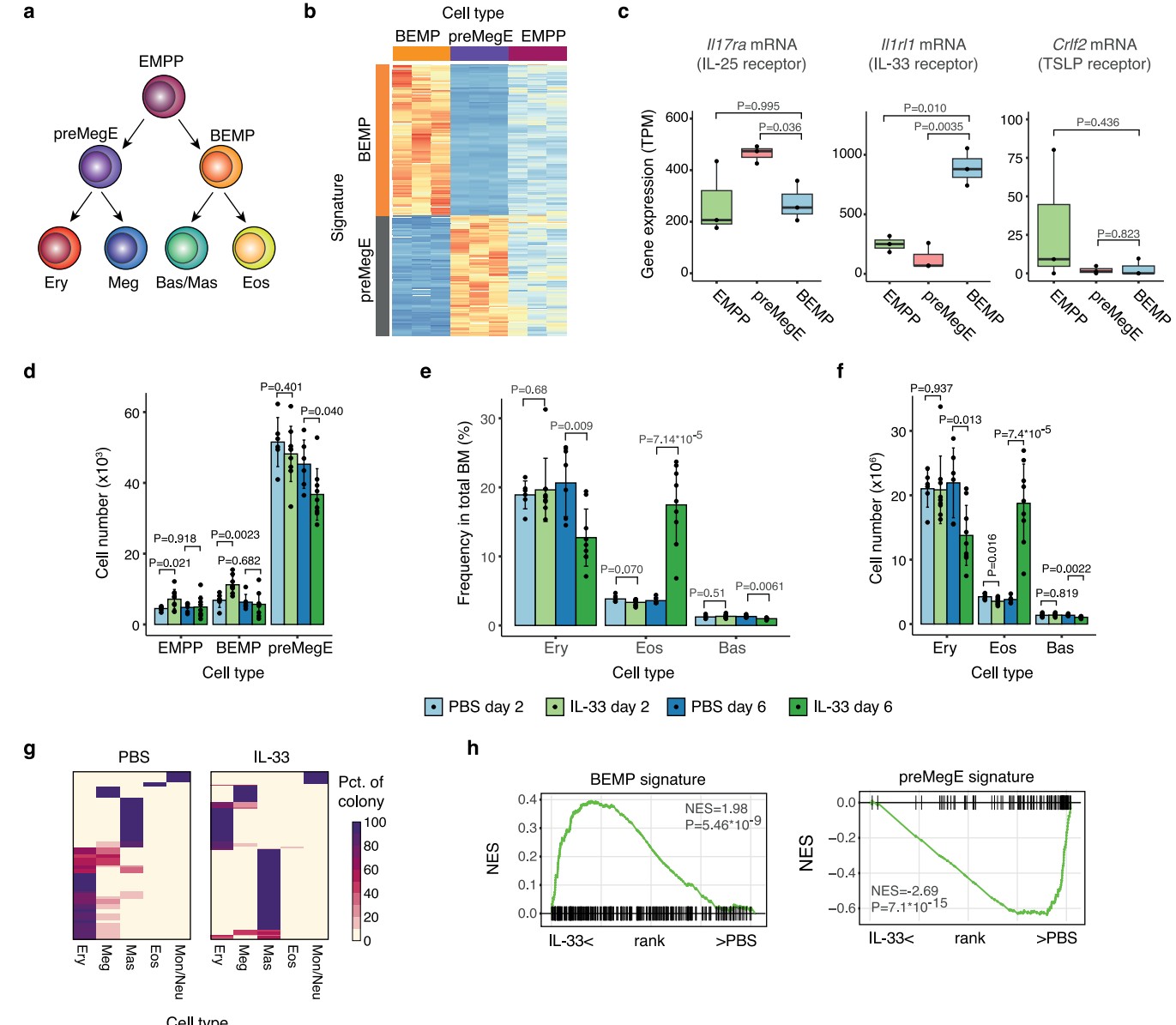

**Extended Data Fig. 2 | IL-33 regulates EMPP lineage output. a**, Schematic of the differentiation hierarchy of *Gata1* expressing progenitors. **b**, Heatmap of gene expression of BEMP and preMegE signatures, defined as genes differentially expressed between BEMPs and preMegEs (Padj<0.05), in EMPPs, BEMPs and preMegEs. **c**, Expression level of Il17ra, Il1rl1 and Crlf2 genes in EMPP, preMegE and BEMP cell population measured by RNA-seq; 3 biological replicates. **d**, Quantification of EMPP, BEMP and preMegE cell number isolated from the bone marrow of mice, related to Fig. 1h,i. **e**, Quantification of erythroid (Ery), eosinophils (Eos) and basophils (Bas) cell number isolated from the bone marrow of mice related to Fig. 1h,i. **f**, Frequency of Ery, Eos and Bas within the *Gata1* expressing cell fraction in BM from (f). PBS day 2, n = 6; PBS day 6, n = 6;

IL-33 day 2, n = 9; IL-33 day 6, n = 9. 3 independent experiments. **g**, Clonal fate heatmap showing the cell type frequency in colonies generated by EMPP isolated from mice treated with IL-33 (n = 105 colonies) or PBS (n = 91 colonies) for 2 days. Ery, Meg, Mas, Eos and Mon/Neu frequencies were quantified by flow cytometry. 6 independent experiments. **h**, GSEA analysis comparing the expression of the BEMP (left panel) and preMegE gene expression signatures (right panel) between EMPPs isolated from mice treated with IL-33 or PBS for 2 days using RNA-seq (n = 4/condition). NES and p-values are shown for each comparison. Values are mean ± SD. P-values were determined by two-tailed Welch's t-test unless otherwise specified.

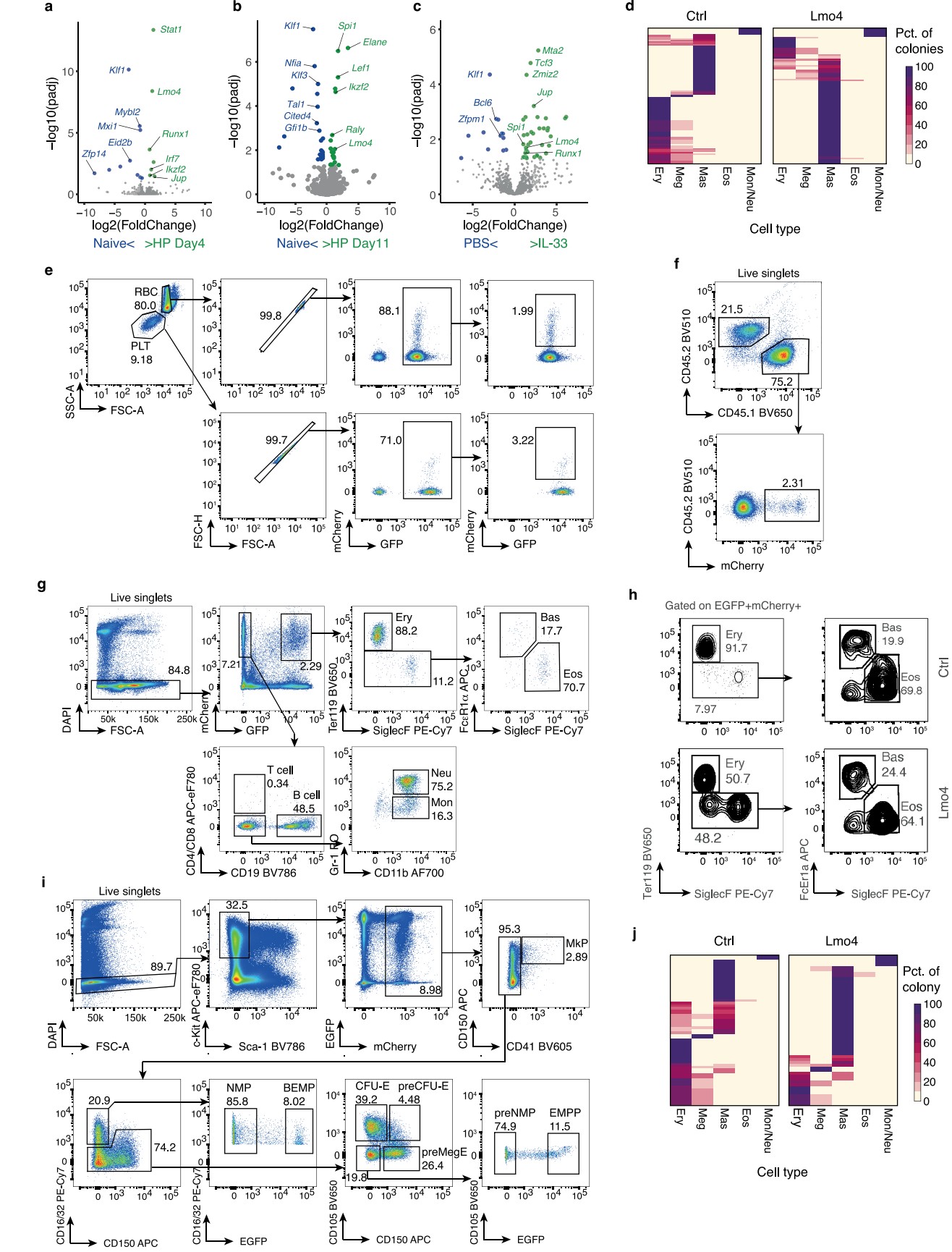

**Extended Data Fig. 3** | See next page for caption.

**Extended Data Fig. 3 | *Lmo4* is induced by both *H. Polygyrus* and IL-33.**
**a**, Differential transcription factor (TF) gene expression between EMPP purified from naïve mice and H.polygyrus infected mice (day 4). n = 4/condition. **b**, Differential transcription factor (TF) gene expression between EMPP purified from naïve mice and H.polygyrus infected mice (day 11). n = 4/condition. **c**, Differential transcription factor (TF) gene expression between EMPP purified from mice injected with IL-33 and control PBS-injected mice (day 2). n = 3/condition. **d**, Clonal fate heatmap showing the cell type frequency colonies generated by EMPP transduced ex vivo with *Lmo4*-expressing (n = 101) or control (n = 99) lentivirus. Ery, Meg, Mas, Eos and Mon/Neu frequencies were quantified by flow cytometry. **e**, Gating strategy used to quantify the frequency of mCherry+ cells in donor derived (EGFP+) red blood cells (RBC) and platelets (PLT) in the peripheral blood of recipient mice in Fig. 2h. **f**, Gating strategy used to quantify the frequency of mCherry+ cells in donor derived mononuclear cells (CD45.1+), in the peripheral blood of recipient mice in Fig. 2h. **g**, Flow cytometry workflow for quantification of *Gata1*+ blood cell types within the mCherry+ BM cell fraction in mice from Fig. 2i. The number of cells is shown as percentage of the parental gate. **h**, Representative example of quantification of Lmo4 transduced and control (Ctrl) erythroid (Ery), eosinophil (Eos) and basophil (Bas) cells from recipient mice represented in Fig. 2i. **i**, Gating strategy used to quantify the frequency of EMPP, preMegE and BEMP cells in the Lin−Sca-1−c-Kit+mCherry+ cell fraction in BM of recipient mice in Fig. 2j. **j**, Clonal fate heatmap showing the cell type frequency in colonies generated by mCherry+ EMPP isolated from mice transplanted as shown in Fig. 2g, *Lmo4* (n = 54) or control transduced (n = 61). Cell types were quantified by flow cytometry. 3 independent experiments. Differential gene expression P-values were computed using a two-sided Wilcoxon Rank-Sum test with Bonferroni correction (a-c).

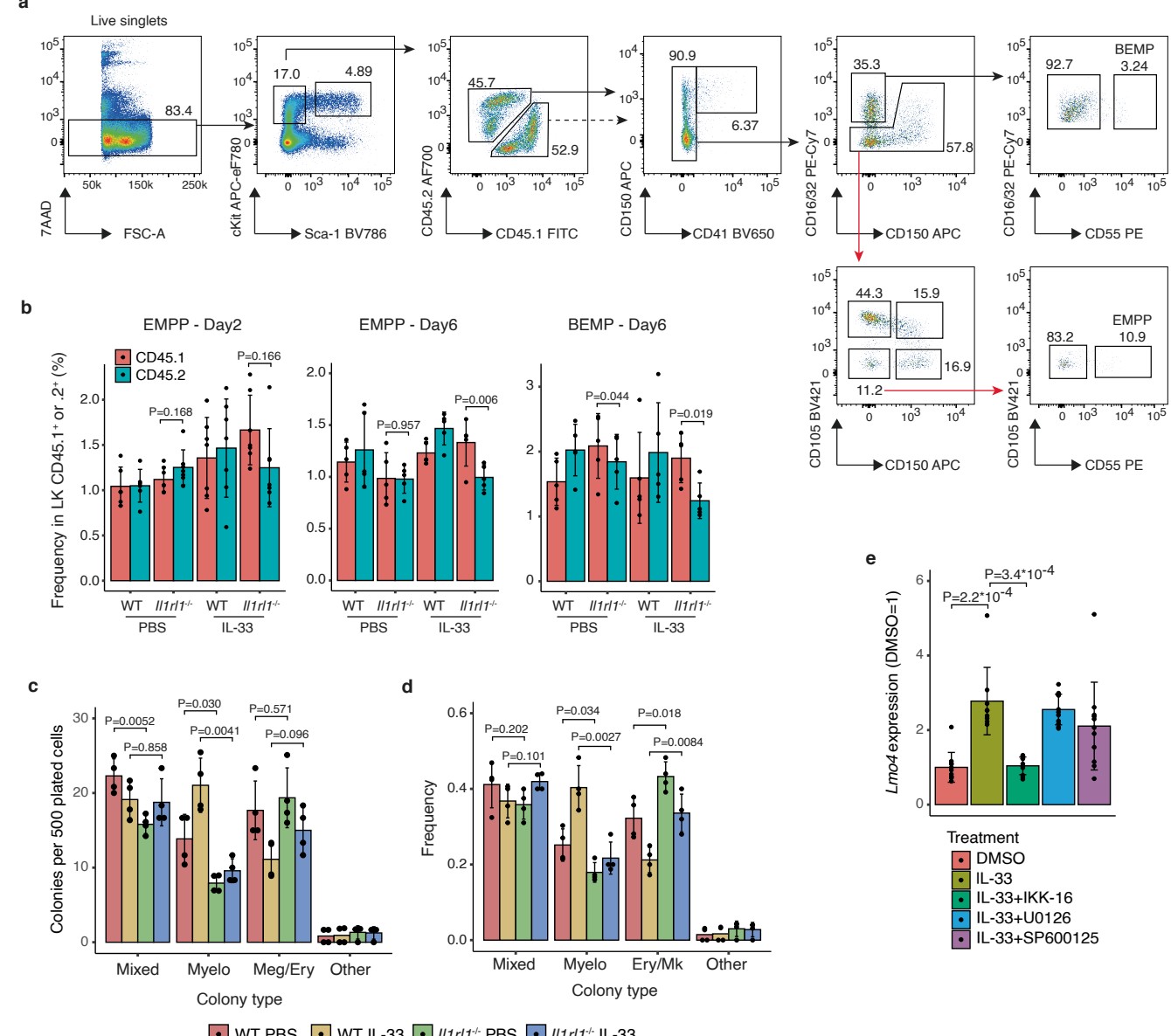

**Extended Data Fig. 4 | IL-33 acts on EMPPs to control their lineage bias.**
**a**, Gating strategy used to quantify the frequency of EMPP, BEMP and preMegE with the LK CD45.1+ and LK CD45.2+ population, related to Fig. 3a,b. **b**, Frequency of EMPP and BEMP progenitors within LK CD45.1+ or LK CD45.2+ populations in *Il1rl1-/-* bone marrow chimeras treated with mouse recombinant IL-33 or PBS for 2 or 6 consecutive days as shown in Fig. 3a. For the day 2 timepoint, n = 7 for each condition except WT PBS (n = 6); for the day 6 timepoint, n = 5 per condition. Indicated P-values have been calculated comparing the number of CD45.1+ and CD45.2+ BEMPs within each mouse using a paired Student's t-test. **c**, Quantification of clonal EMPP fates as number of colonies per 500

plated cells from the indicated condition as illustrated in Fig. 3a classified as follows: Mixed: Ery and/or Meg with Mas and/or Eos; Myelo: Mas and/or Eos; Ery/Meg: Ery and/or Meg colonies. Other: Mon/Neu **d**, Clonal EMPP fates from (c) as frequencies of total colonies. 4 independent experiments. **e**, *Lmo4* gene expression in EMPP populations isolated from *Gata1*-EGFP mice, treated for 16 h with IL-33 and inhibitors as indicated. DMSO (n = 11), IL-33 (n = 9), IL-33/ IKK-16 (n = 10), IL-33/SP600125 (n = 11) and IL-33/U0126 (n = 11). 2 independent experiments. Values are mean ± SD. Statistical significance was determined by two-tailed Welch's t-test.

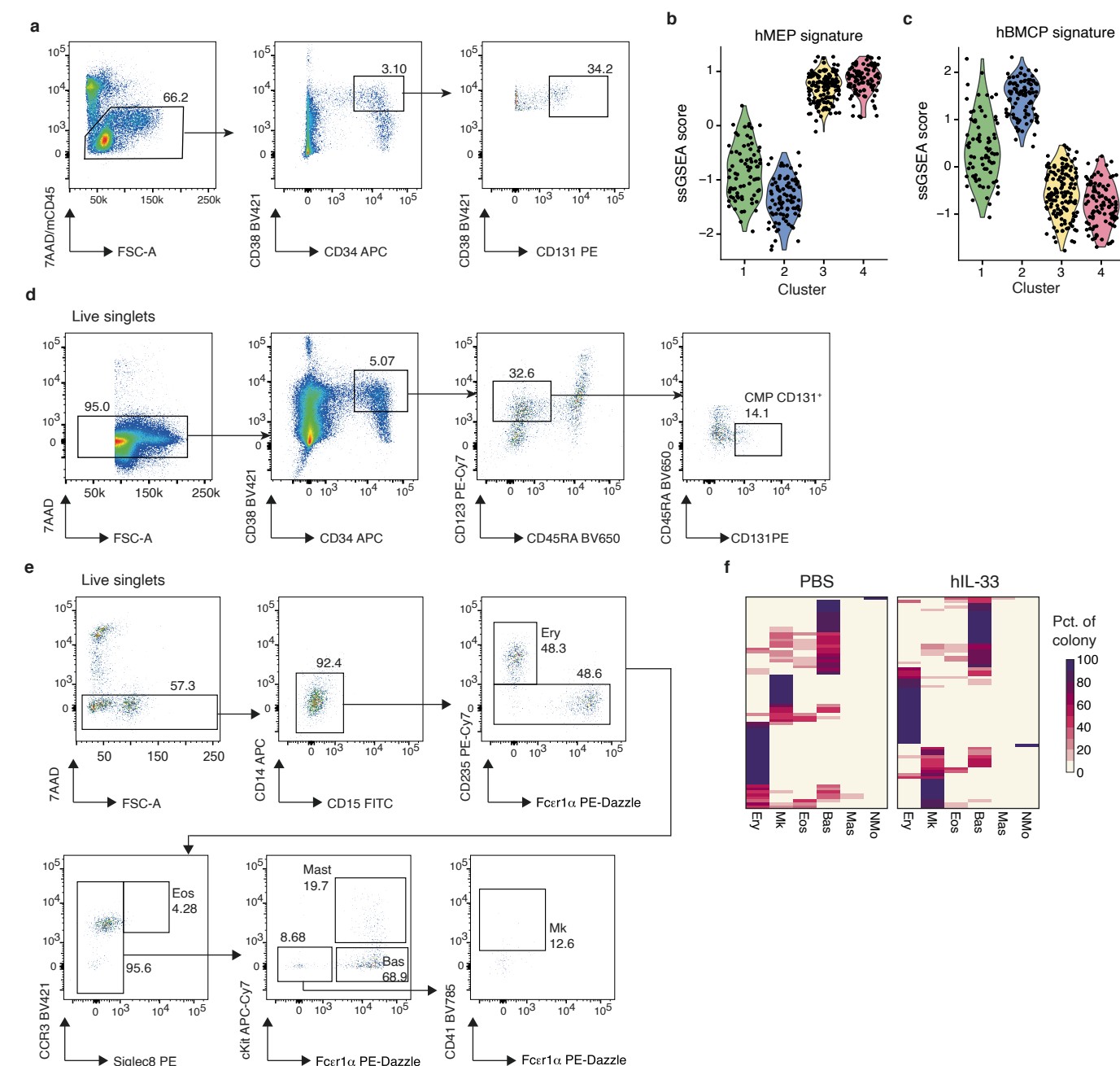

**Extended Data Fig. 5 | Effect of hIL-33 on human EMPPs. a**, Gating strategy used to sort human CD34+CD38−CD131+ cells from bone marrow of xenografted mice, related to Fig. 3f–h. **b**, Histogram showing the human MEP (hMEP) ssGSEA signature score of the cell clusters from Fig. 3f. **c**, Histogram showing the human BMCP (hBMCP) ssGSEA signature score of the cell clusters from Fig. 3f. **d**, Gating strategy used to sort human CMP CD131+ cells, related to Fig. 3i. **e**, Gating strategy used to quantify lineage clonal fate from human CMP CD131+ cell, related to Fig. 3i. **f**, Clonal fate heatmap showing the cell type frequency colonies generated by human CMP CD131+ cell treated with recombinant human IL-33 or PBS, related to Fig. 3i. Ery, Mk, Mas, Eos, Bas and Mon/Neu frequencies were quantified by flow cytometry as in (e).

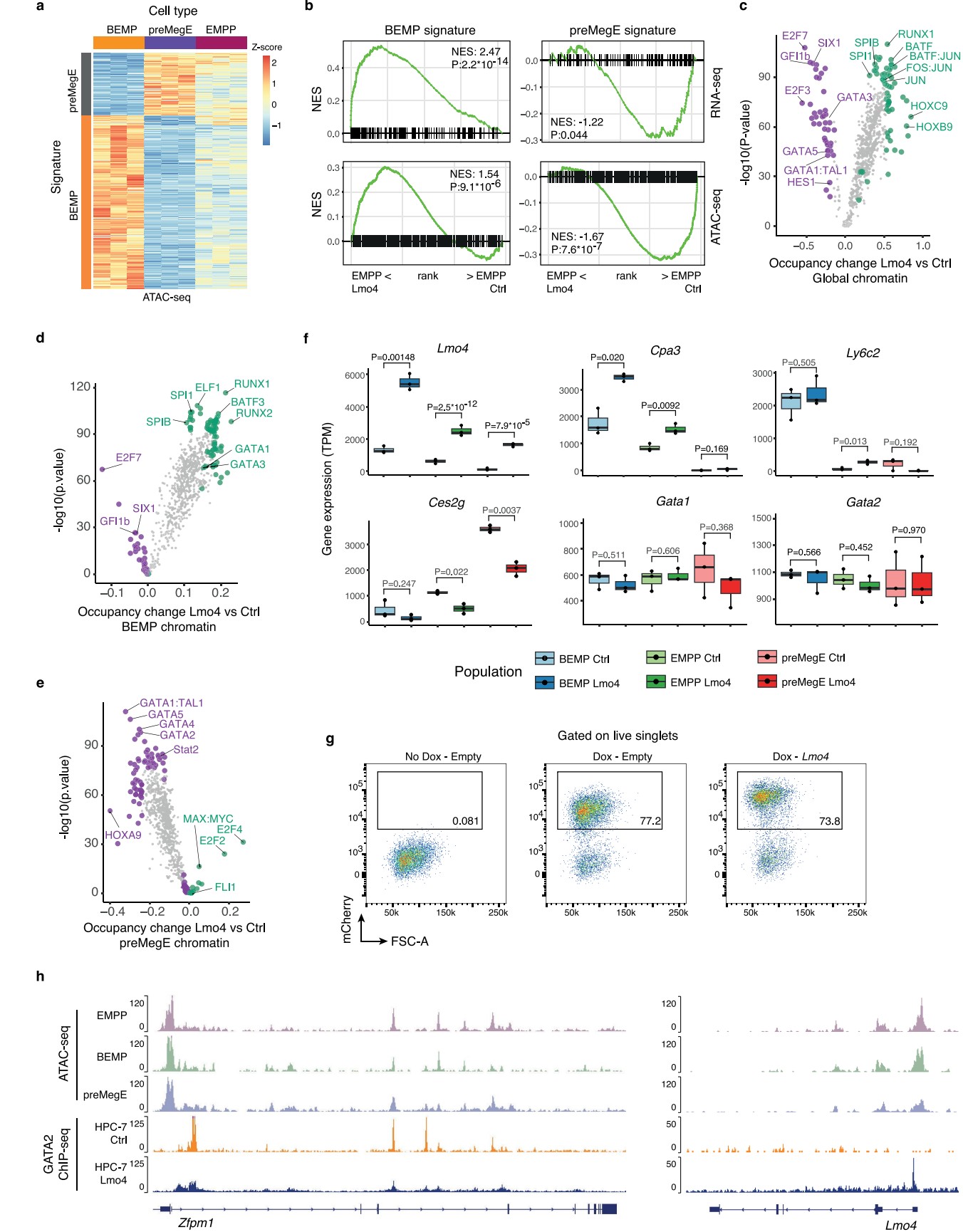

**Extended Data Fig. 6** | See next page for caption.

**Extended Data Fig. 6 | Lmo4 reprograms EMPPs. a**, Heatmap of chromatin accessibility in EMPP, BEMP and preMegE of BEMP (orange) and preMegE (grey) signatures identified as peaks differentially accessible (Padj<0.05) between BEMPs and preMegEs. **b**, GSEA analysis of BEMP and preMegE gene expression (RNA-seq) and chromatin accessibility signatures (ATAC-seq) comparing *Lmo4* and control virus transduced mCherry+ EMPPs from mice transplanted as in Fig. 2g. n = 3/condition. NES and p-values are shown for each comparison. **c**, TF motifs differentially occupied between *Lmo4* and control virus transduced mCherry+ EMPPs isolated from mice transplanted as shown in Fig. 2g identified using TOBIAS. n = 3/condition. Large dots represent differentially bound TFs and colours indicate increased binding in *Lmo4* (green) or in Ctrl (purple) conditions, respectively. **d**, TF motifs differentially occupied within BEMP-specific chromatin between LMO4 transduced and control mCherry+ EMPPs identified as in (c). **e**, TF motifs differentially occupied within preMegE-specific chromatin between LMO4 transduced and control mCherry+ EMPPs identified as in (c). **f**, Expression level (TPM) of *Lmo4*, *Cpa3*, *Ly6c2*, *Ces2g*, *Gata1* and *Gata2* genes in mCherry+ BEMP, EMPP and preMegE cells purified from recipient mice transplanted as shown in Fig. 2g (n = 3/condition). P-values were computed using a two-sided Wilcoxon Rank-Sum test with Bonferroni correction (a-c). **g**, Gating strategy used to sort HPC-7 cells ectopically expressing *Lmo4* or empty backbone (Ctrl) using a doxycycline (Dox) inducible lentiviral vector. No Dox control is also shown. **h**, Genome browser tracks showing ATAC-seq and GATA-2 ChIP-seq signals in the indicated cell population for the *Zfpm1* and *Lmo4* gene loci. P-values for differential TF motif occupancy (c-e) were computed using TOBIAS.

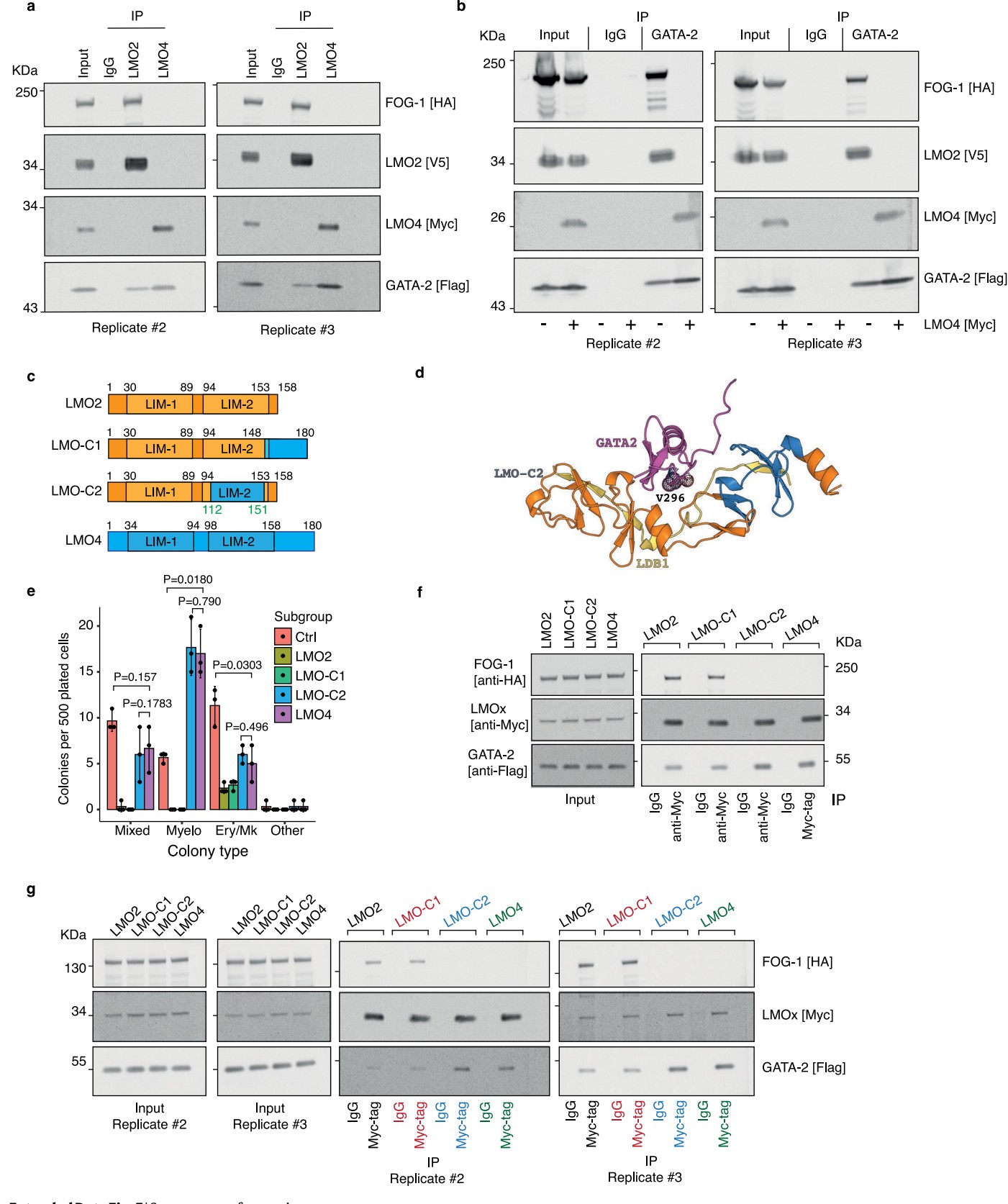

**Extended Data Fig. 7** | See next page for caption.

**Extended Data Fig. 7 | Analysis of GATA-2–Lmo4 interaction. a**, Replicates of co-immunoprecipitation assay from Fig. 4b. **b**, Replicates of co-immunoprecipitation assay from Fig. 4c. **c**, Schematic of LMO2, LMO4, and the LMO-C1 and LMO-C2 chimeras. Domain start and end positions are indicated. **d**, The structure and binding interface of the LMO-C2 construct (residues 112-151 from LMO4 highlighted in blue), LDB1 LID (337-375) (yellow) and GATA-2 NF (light magenta) as modelled by Alphafold3. GATA-2 V296 is highlighted with a mesh surface. **e**, Clonal fates of isolated *Gata1*-EGFP EMPPs transduced with lentivirus expressing the indicated LMO constructs. Mixed: Ery and/or Meg with Mas and/or Eos; Myelo: Mas and/or Eos; Ery/Meg: Ery and/or Meg colonies. Other: Mon/Neu. 500 cells/assay, 3 biological replicates. **f**, HA-FOG-1, Flag-GATA-2 and Myc-LMO were co-expressed in HEK293T cells and co-immunoprecipitated with anti-Myc antibodies or control IgG followed by Western blotting with the indicated anti-tag antibodies. The input represents 0.75% of the material used for immunoprecipitation. **g**, Replicates of co-immunoprecipitation assay from (f).

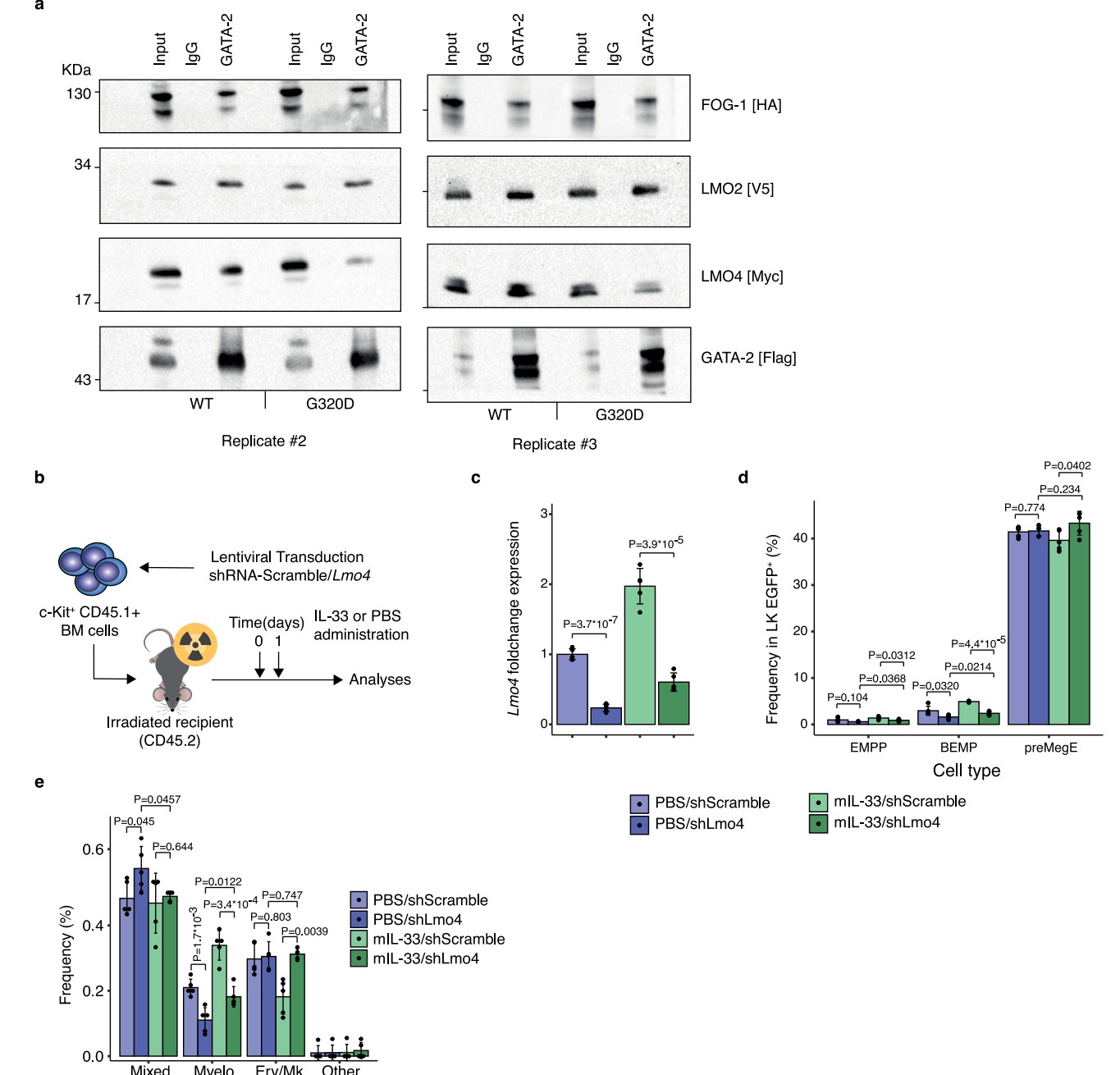

**Extended Data Fig. 8 | Lmo4 is required for myeloid commitment.**
**a**, Replicates of co-immunoprecipitation assay from Fig. 4f. **b**, Experimental workflow schematic showing the transduction of c-Kit+ isolated from 9-week old C57BL/6 mice with lentivirus expressing shRNA targeting Lmo4 or Scramble, and transplanted into lethally irradiated 10-week old C57BL/6 female mice. Treatment with recombinant mouse IL-33 or PBS for 2 consecutive days is also represented. **c**, Quantification of *Lmo4* gene expression in EMPP isolated from transplanted and treated mice as shown in h). n = 5 biological replicate per condition. **d**, Frequency of EMPP, BEMP and preMegE cell populations within the LK BM cell fraction in transplanted and treated mice as shown in (a). 5 biological replicates. **e**, Quantification of clonal EMPP fates isolated from transplanted and treated mice as shown in h). n = 5 biological replicate per condition. Mixed: Ery and/or Meg with Mas and/or Eos; Myelo: Mas and/or Eos; Ery/Meg: Ery and/or Meg colonies. Other: Mon/Neu. Values are mean ± SD. P-values were determined by two-tailed Welch's t-test.

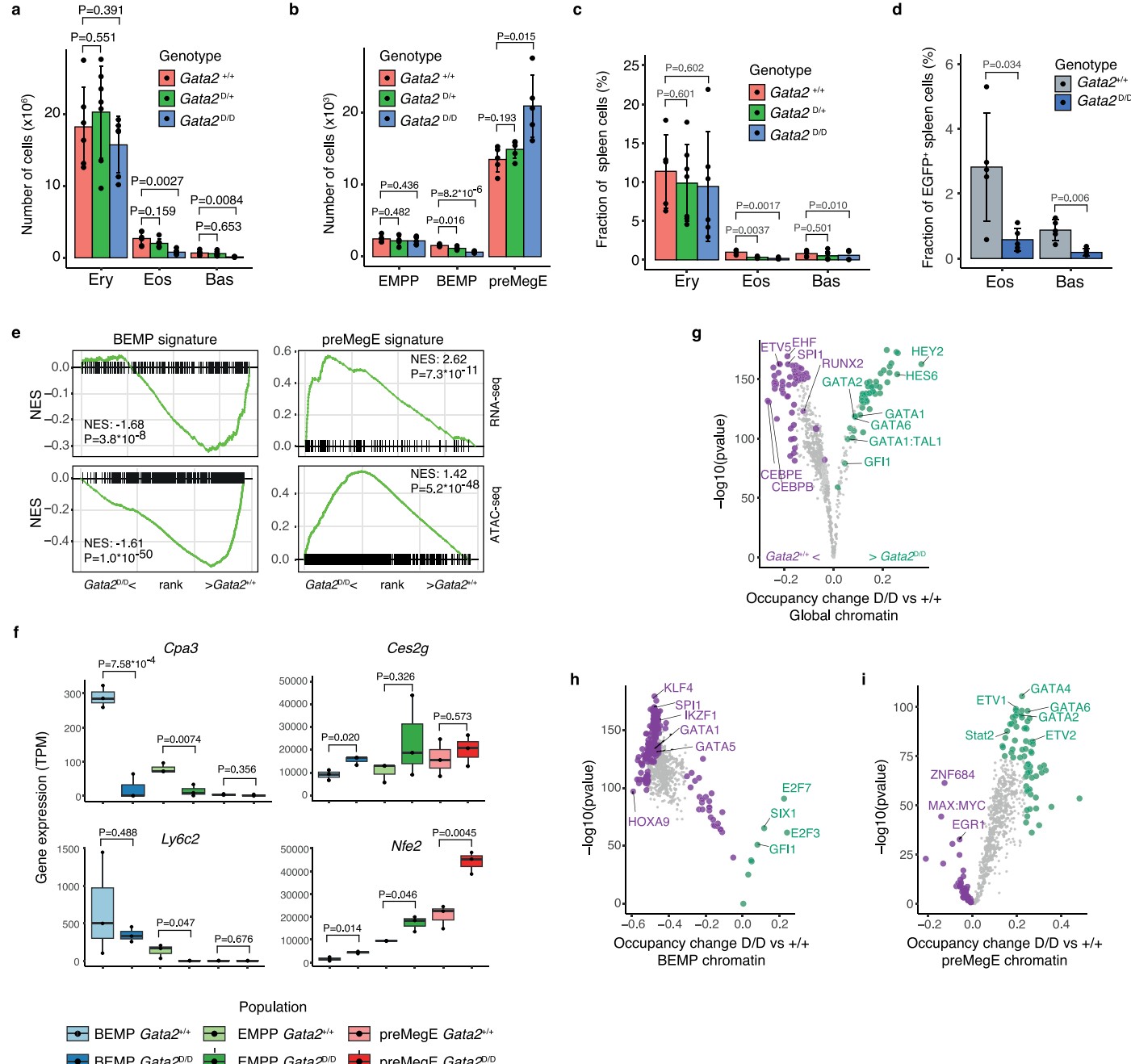

**Extended Data Fig. 9 | GATA-2–Lmo4 interaction controls EMPP lineage bias. a**, Quantification of erythroid (Ery), eosinophils (Eos) and basophils (Bas) in the bone marrow of *Gata2*[+/+] (n = 6), *Gata2*[D/+] (n = 7) and *Gata2*[D/D] (n = 6) mice. Related to Fig. 5a. **b**, Quantification of EMPP, BEMP and preMegE cell number isolated from the bone marrow of mice, related to Fig. 5b. n = 5 per genotype. **c**, Frequency of erythroid (Ery), eosinophil (Eos) and basophil (Bas) cells in the spleen of *Gata1*-EGFP mice with a *Gata2*[+/+] (n = 6), *Gata2*[D/+] (n = 7) and *Gata2*[D/D] (n = 6) genotype measured by flow cytometry. Values are mean frequencies ±SD from 3 independent experiments. P-values are shown. **d**, Frequency of eosinophils (Eos) and basophils (Bas) in the EGFP+ spleen cell fraction of mice from Fig. 5c. n = 9/genotype. **e**, GSEA analysis of BEMP and preMegE gene expression (RNA-seq) and chromatin accessibility signatures (ATAC-seq) comparing *Gata2*[+/+] and *Gata2*[D/D] EMPPs. n = 3/condition. NES and p-values are shown for each comparison. **f**, Expression level (TPM) of *Nfe2*,

*Cpa3*, *Ly6c2* and *Ces2g* in BEMP, EMPP or preMegE cells isolated from *Gata1*-EGFP mice with a *Gata2*[+/+] (n = 3), or *Gata2*[D/D] (n = 3) genotype measured by RNA-seq. Values are mean ± SD from 3 independent experiments. P-values were computed using a two-sided Wilcoxon Rank-Sum test with Bonferroni correction. **g**, TF motifs differentially occupied between EMPPs with a *Gata2*[+/+] or *Gata2*[D/D] genotype isolated from mice transplanted as in Fig. 5c identified using TOBIAS. Large dots indicate motifs showing significant binding differences. **h**, TF motifs differentially occupied within BEMP-specific chromatin in EMPPs with a *Gata2*[+/+] or *Gata2*[D/D] genotype identified as in (g). **i**, TF motifs differentially occupied within preMegE-specific chromatin in EMPPs with a *Gata2*[+/+] or *Gata2*[D/D] genotype identified as in (g). P-values for differential TF motif occupancy (c-e) were computed using TOBIAS. Statistical significance was determined by two-tailed Welch's t-test except when otherwise specified.

# Reporting Summary

## Statistics

For all statistical analyses, confirm that the following items are present in the figure legend, table legend, main text, or Methods section.

| n/a | Confirmed | |
|---|---|---|
| ☐ | ☒ | The exact sample size (*n*) for each experimental group/condition, given as a discrete number and unit of measurement |
| ☐ | ☒ | A statement on whether measurements were taken from distinct samples or whether the same sample was measured repeatedly |
| ☐ | ☒ | The statistical test(s) used AND whether they are one- or two-sided<br>*Only common tests should be described solely by name; describe more complex techniques in the Methods section.* |
| ☐ | ☒ | A description of all covariates tested |
| ☐ | ☒ | A description of any assumptions or corrections, such as tests of normality and adjustment for multiple comparisons |
| ☐ | ☒ | A full description of the statistical parameters including central tendency (e.g. means) or other basic estimates (e.g. regression coefficient) AND variation (e.g. standard deviation) or associated estimates of uncertainty (e.g. confidence intervals) |
| ☐ | ☒ | For null hypothesis testing, the test statistic (e.g. *F*, *t*, *r*) with confidence intervals, effect sizes, degrees of freedom and *P* value noted<br>*Give P values as exact values whenever suitable.* |
| ☒ | ☐ | For Bayesian analysis, information on the choice of priors and Markov chain Monte Carlo settings |
| ☒ | ☐ | For hierarchical and complex designs, identification of the appropriate level for tests and full reporting of outcomes |
| ☒ | ☐ | Estimates of effect sizes (e.g. Cohen's *d*, Pearson's *r*), indicating how they were calculated |

*Our web collection on statistics for biologists contains articles on many of the points above.*

## Software and code

Policy information about availability of computer code

| Data collection | FACS data were collected on BD FACSAria III, BD FACSAria Fusion or BD FACS Fortessa X-20 using FACS DIVA software (v9.0) |
|---|---|
| Data analysis | FACS data was analysed using Flowio software (version 10.8.1) and subsequent statistical analyses performed using R (v4.2.2) RNA-seq, ATAC-seq and ChIP-seq were analysed as described in the method section, using the following open source software and R packages: Trim Galore ! (version 0.6.5), Salmon (version 1.2.0), DESeq2(version 1.42.0), java GSEAversion 4.2.3), FGSEA(version 1.34.2 http://bioconductor.org/packages/fgsea/), Bowtie2 (version 2.4.1), SAMtools (version 1.17), MACS2 (version 2.1.2), bamCoverage (deepTools 3.5.3), featuresCounts (version 2.0.1) and TOBIAS (version 0.12.1). Single-cell gene expression analysis was performed as described in the method section, using Seurat v5 and ssGSEA (https://rpubs.com/pranali018/SSGSEA). Structural modelling was performed using AlphaFold3 (2024) and visualized using PyMOL Molecular Graphics System (version 2.1.0). |

For manuscripts utilizing custom algorithms or software that are central to the research but not yet described in published literature, software must be made available to editors and reviewers. We strongly encourage code deposition in a community repository (e.g. GitHub). See the Nature Portfolio guidelines for submitting code & software for further information.

# Data

Policy information about availability of data

All manuscripts must include a data availability statement. This statement should provide the following information, where applicable:
- Accession codes, unique identifiers, or web links for publicly available datasets
- A description of any restrictions on data availability
- For clinical datasets or third party data, please ensure that the statement adheres to our policy

Raw data and proceed data files generated by sequencing are available at Gene Expression Omnibus (GEO) as part of the super-series GSE276830. mm10 (GRCm38) and hg19 (GRCh37) reference genome sequences can be obtained from Gencode (https://www.gencodegenes.org). No homemade code has been used in this study and could be provided upon reasonable request.

# Research involving human participants, their data, or biological material

Policy information about studies with human participants or human data. See also policy information about sex, gender (identity/presentation), and sexual orientation and race, ethnicity and racism.

| | |
|---|---|
| Reporting on sex and gender | *Use the terms sex (biological attribute) and gender (shaped by social and cultural circumstances) carefully in order to avoid confusing both terms. Indicate if findings apply to only one sex or gender; describe whether sex and gender were considered in study design; whether sex and/or gender was determined based on self-reporting or assigned and methods used.*<br>*Provide in the source data disaggregated sex and gender data, where this information has been collected, and if consent has been obtained for sharing of individual-level data; provide overall numbers in this Reporting Summary. Please state if this information has not been collected.*<br>*Report sex- and gender-based analyses where performed, justify reasons for lack of sex- and gender-based analysis.* |
| Reporting on race, ethnicity, or other socially relevant groupings | *Please specify the socially constructed or socially relevant categorization variable(s) used in your manuscript and explain why they were used. Please note that such variables should not be used as proxies for other socially constructed/relevant variables (for example, race or ethnicity should not be used as a proxy for socioeconomic status).*<br>*Provide clear definitions of the relevant terms used, how they were provided (by the participants/respondents, the researchers, or third parties), and the method(s) used to classify people into the different categories (e.g. self-report, census or administrative data, social media data, etc.)*<br>*Please provide details about how you controlled for confounding variables in your analyses.* |
| Population characteristics | *Describe the covariate-relevant population characteristics of the human research participants (e.g. age, genotypic information, past and current diagnosis and treatment categories). If you filled out the behavioural & social sciences study design questions and have nothing to add here, write "See above."* |
| Recruitment | *Describe how participants were recruited. Outline any potential self-selection bias or other biases that may be present and how these are likely to impact results.* |
| Ethics oversight | *Identify the organization(s) that approved the study protocol.* |

Note that full information on the approval of the study protocol must also be provided in the manuscript.

# Field-specific reporting

Please select the one below that is the best fit for your research. If you are not sure, read the appropriate sections before making your selection.

☒ Life sciences    ☐ Behavioural & social sciences    ☐ Ecological, evolutionary & environmental sciences

For a reference copy of the document with all sections, see nature.com/documents/nr-reporting-summary-flat.pdf

# Life sciences study design

All studies must disclose on these points even when the disclosure is negative.

| | |
|---|---|
| Sample size | Sample sizes were chosen based on the variance previously observed in similar experiments, to allow detection of a 50% difference at P=0.05 with 80% power. Block designs were used to allow variance estimates to be adjusted if necessary |
| Data exclusions | No data was excluded in this study |
| Replication | The number of replicate for each experiment is indicated in the figure legend of the manuscript. For RNA-seq, each biological replicate was isolated from individual animal. For ATAC-seq, each biological replicate was generated from a pool of 3 animal. For each experiment, high-throughput sequencing libraries were sequenced in a single library pool to reduce batch effect. For in vitro and in vivo assays, experiment were performed at least twice independently. |
| Randomization | For in vivo experiment, sex and age-matched animal mice were randomly selected and allocated to experimental groups (i.e: PBS vs IL-33). For other experiment, randomization was not relevant since experimental groups were assigned based on mouse genotype or previous |

treatment.

Blinding | For in vivo experiment, data collection was performed blinded and experimental groups allocated to respective samples at the end of the analysis. For in vitro assays, experimental groups were required to be known during data collection for correct repartition. No analyses required manual counting or subjective analysis were performed and thus not subject to investigator biais.

# Reporting for specific materials, systems and methods

We require information from authors about some types of materials, experimental systems and methods used in many studies. Here, indicate whether each material, system or method listed is relevant to your study. If you are not sure if a list item applies to your research, read the appropriate section before selecting a response.

## Materials & experimental systems

| n/a | Involved in the study |
|---|---|
| ☐ | ☒ Antibodies |
| ☐ | ☒ Eukaryotic cell lines |
| ☒ | ☐ Palaeontology and archaeology |
| ☐ | ☒ Animals and other organisms |
| ☒ | ☐ Clinical data |
| ☒ | ☐ Dual use research of concern |
| ☒ | ☐ Plants |

## Methods

| n/a | Involved in the study |
|---|---|
| ☐ | ☒ ChIP-seq |
| ☐ | ☒ Flow cytometry |
| ☒ | ☐ MRI-based neuroimaging |

## Antibodies

Antibodies used

SiglecF E50-2440 PE 552126 BD Biosciences
CD150 TC15-12F12.2 APC 115910 Biolegend
FceR1a mars-01 APC 17-5898-82 eBioscience
c-kit 2B8 APC-eFluor780 47-1171-82 eBioscience
CD4 RM4-5 APC-eFluor780 47-0042-82 eBioscience
CD8a 53-6.7 APC-eFluor780 47-0081-82 eBioscience
CD11b M1/70 Alexa Fluor 700 56-0112-80 eBioscience
CD16/32 93 PE-Cy7 25-0161-82 eBioscience
CD19 1D3 PE-Cy7 25-0193-81 eBioscience
CD41 MWRAG30 BV421 133911 Biolegend
SiglecF E50-2440 BV421 562681 BD Biosciences
CD105 MJ7/18 BV421 562760 BD Biosciences
Ter119 TER-119 BV650 116235 Biolegend
CD41 MWReg30 BV650 740504 BD Biosciences
CD105 MJ7/18 BV650 740609 BD Biosciences
Sca-1 D7 BV786 563991 BD Biosciences
Ly-6G/Ly-6C RB6-8C5 Pacific Orange RM3030 Invitrogen
Lmo4 EPR6731(2) NA ab131030 Abcam
Streptavidin NA PE 12-4317-87 eBioscience
CD45.1 A20 FITC 11-0453-85 eBioscience
CD45.2 104 AF700 56-0454-82 eBioscience
CD14 63D3 APC 367118 Biolegend
CD15 W6D3 FITC 323004 Biolegend
CD117 104D2 APC-Fire750 313240 Biolegend
CCR3 5.00E+08 BV421 310714 Biolegend
Fcer1a AER-37 PE-Dazzle 334634 Biolegend
Siglec8 7C9 PE 347104 Biolegend
CD235a HI264 PE-Cy7 349112 Biolegend
CD41 HIP8 BV785 303744 Biolegend
CD34 581 APC 343510 Biolegend
CD38 HIT2 BV421 303526 Biolegend
CD123 6H6  PE-Cy7 306010 Biolegend
CD45RA HI100  BV650 304136 Biolegend
CD131 1C1 PE 306104 Biolegend
anti-GATA2 AF2046 N/A AF2046 R&D
anti-Flag  M2 N/A F3165 Sigma
anti-Flag  2368 N/A 2368S Cell Signaling
anti-Myc  71D10 N/A 2278 Cell Signaling
anti-Myc  9.00E+10 N/A MA1-980 Invitrogen
anti-HA 2-2.2.14 N/A 26183 Invitrogen
anti-V5 A190-120A N/A A190-120A Bethyl Laboratories

Validation | Individual antibodies have been titrated to identify the optimal concentration to be used, as previously described (Drissen et al., 2016, Meng et al., 2023)

# Eukaryotic cell lines

Policy information about cell lines and Sex and Gender in Research

| | |
|---|---|
| Cell line source(s) | HEK-293T cell line (Sigma, 12022001-DNA-5UG)<br>HPC-7 cell line (Pinto do O et al., 1998 ; Wilson K.N et al., 2010) was a kind gift from Pr. de Bruijn MFTR. |
| Authentication | HEK-193T was purchased from the supplier (Sigma) as an authenticated cell line (by STR profiling). HPC-7 cell line was authenticated by differentiation into erythroid, megakaryocyte and myeloid lineages (18 July 2024), as previously described. |
| Mycoplasma contamination | Cell lines were tested negative for mycoplasm contamination. |
| Commonly misidentified lines<br>(See ICLAC register) | No commonly misidentified cells were used |

# Animals and other research organisms

Policy information about studies involving animals; ARRIVE guidelines recommended for reporting animal research, and Sex and Gender in Research

| | |
|---|---|
| Laboratory animals | 8-16 week old mice were used in this study. Mouse strain used are listed bellow:<br>- C57BL/6 CD45.1 (Jackson 002014)<br>- Gata1-EGFP (BAC transgenic; Drissen et al. 2016).<br>- ST2ko (Jackson 039982; Townsend et al., 2000).<br>- NOD.Cg-KitW-41J Tyr+ Prkdcscid Il2rgtm1Wjl/ThomJ (NSGW41; McIntosh B.E., et al, 2015).<br><br>Mice were housed on a 12h light-dark cycle in a standard micro-isolator cages with 45-65% environmental humidity and 19-23°C temperature. Enrichment consisting of red translucent plastic house and crinkle cut naturalistic paper strands. |
| Wild animals | No wild animals were used |
| Reporting on sex | Bone marrow transplantation was performed using female recipient mice. |
| Field-collected samples | This study did not include field-collected samples |
| Ethics oversight | All animal studies were performed in accordance with the UK Home Office regulations with approval by the University of Oxford Animal Welfare and Ethical Review Body (project license number 30/3359, PP2240412 and PP3246892) and the Francis Crick Institute Animal Welfare and Ethical Review Body (project license number PP8468807), as well as the Irish Health Products Regulatory Authority regulations Licence AE19136/P108 and P196, approved by Trinity College Dublin's Animal Research Ethics Committee |

Note that full information on the approval of the study protocol must also be provided in the manuscript.

# Plants

| | |
|---|---|
| Seed stocks | *Report on the source of all seed stocks or other plant material used. If applicable, state the seed stock centre and catalogue number. If plant specimens were collected from the field, describe the collection location, date and sampling procedures.* |
| Novel plant genotypes | *Describe the methods by which all novel plant genotypes were produced. This includes those generated by transgenic approaches, gene editing, chemical/radiation-based mutagenesis and hybridization. For transgenic lines, describe the transformation method, the number of independent lines analyzed and the generation upon which experiments were performed. For gene-edited lines, describe the editor used, the endogenous sequence targeted for editing, the targeting guide RNA sequence (if applicable) and how the editor was applied.* |
| Authentication | *Describe any authentication procedures for each seed stock used or novel genotype generated. Describe any experiments used to assess the effect of a mutation and, where applicable, how potential secondary effects (e.g. second site T-DNA insertions, mosiacism, off-target gene editing) were examined.* |

# ChIP-seq

## Data deposition

☒ Confirm that both raw and final processed data have been deposited in a public database such as GEO.

☒ Confirm that you have deposited or provided access to graph files (e.g. BED files) for the called peaks.

| | |
|---|---|
| Data access links<br>*May remain private before publication.* | Raw data and proceed data files generated by sequencing are available at Gene Expression Omnibus (GEO) as part of the super-series GSE276830 |
| Files in database submission | Both FASTQ and BED files with raw count are provided |

| | |
|---|---|
| Genome browser session<br>(e.g. UCSC) | n/a |

## Methodology

| | |
|---|---|
| Replicates | 2 (HEK-293T dataset) or 3 (HPC-7 dataset) biological replicates have been performed per condition. |
| Sequencing depth | Sequencing depth was 20-40M reads per samples. HEK-293T samples were paired-end and HPC-7 samples were single-end. |
| Antibodies | Antibodies, including supplier reference number, are listed in supplemental Table 12. |
| Peak calling parameters | Peak calling was performed using MACS2 (version 2.1.2) with the following parameters: -q 0.05 --nomodel --extsize 200. |
| Data quality | QC was performed using Trim Galore! version 0.6.5, using QC output file |
| Software | ChIP-seq was analysed as described in the method section, using the following open source software and Rpackages: Trim Galore ! (version 0.6.5), DESeq2(version 1.42.0), Java GSEA version 4.2.3), FGSEA( http://bioconductor.org/packages/fgsea/), Bowtie2 (version 2.4.1), SAMtools (version 1.17), MACS2 (version 2.1.2), bamCoverage (deepTools 3.5.3) and featuresCounts (version 2.0.1) |

# Flow Cytometry

## Plots

Confirm that:

☒ The axis labels state the marker and fluorochrome used (e.g. CD4-FITC).

☒ The axis scales are clearly visible. Include numbers along axes only for bottom left plot of group (a 'group' is an analysis of identical markers).

☒ All plots are contour plots with outliers or pseudocolor plots.

☒ A numerical value for number of cells or percentage (with statistics) is provided.

## Methodology

| | |
|---|---|
| Sample preparation | Bone marrow and peripheral blood cells were prepared as described in the method section |
| Instrument | Stained cells were analysed on LSR Fortessa or LSR X-20 flow cytometers (BD Biosciences). Cell sorting was carried-out on FACSAria III, FACSAria Fusion or FACSymphony S6 (BD Biosciences). |
| Software | Data was collected using FACS DIVA software (v9.0) and analysed using FlowJo software (version 10.8.1) and R (v.4.2.2). |
| Cell population abundance | Post-sort purity was not feasible for sequencing sample collection as cells were directly sort in lysis buffer.<br>Purity was ensure pre-sort by test-sorting of a relevant cell population. |
| Gating strategy | FSC-A/SSC-A was used for gating nucleated cells. FSC-A/FSC-H was used for gating singlet cells. Further specific gating strategy is shown in Extended Data Figures of the manuscript. |

☒ Tick this box to confirm that a figure exemplifying the gating strategy is provided in the Supplementary Information.

