## [Peer Review File · Nature]

A mechanism to initiate emergency type 2 myelopoiesis

Corresponding Author: Professor Claus Nerlov

Version 0:

Reviewer comments:

Referee #1

(Remarks to the Author)

After exposure to parasitic infection, hosts demonstrate a type 2 innate myeloid immune response with increased production of basophils, eosinophils and mast cells (BEM) through incompletely understood mechanisms. The authors study this response in mice infected with the roundworm *H. polygyrus*. It was previously established that parasite infection causes erythroid-primed multipotent progenitors (EMPPs) to differentiate into BEM progenitors at the expense of pre-megakaryocyte-erythroid progenitors (preMegEs), resulting in enhanced production of BEM and reduced red blood cell and platelet production. The current study shows that this lineage fate switch is caused by parasite-induced release of IL-33, which causes EMPPs to upregulate LMO4, a member of an activating transcription factor complex that also contains GATA1 or GATA2, FOG1 and TAL1 (and LDB1). In meg and erythroid lineages, this complex usually contains LMO2. However, during parasitic infection, IL33 caused the replacement of LMO2 with LMO4, which displaced FOG1 and redirected the complex from Meg-Erythroid genes to BEM-related ones, induced expression of the latter and caused a corresponding shift in the lineage output. The authors note this to be an interesting example of transcription factor re-allocation as a mechanism of lineage commitment.

This paper characterizes an interesting mechanism for lineage specification in response to a clinically relevant physiological stress. The lineages examined are relatively rare, difficult to study and poorly understood. The experiments are technically sound and the paper is clearly written. Suggestions for improvement include:

Major

1. Additional mechanistic insights into the key observations would strengthen the paper. Additional outstanding questions include:

a. How do differences in the structures of LMO2 vs LMO4 modulate distinct interactions with FOG1 and GATA2 G320D (or lack thereof)? It should be possible to investigate this by structural modeling combined with additional mutagenesis.

b. What is the mechanism by which IL-33 induces LMO4?

c. A major function of the GATA-TAL-LMO2-LBD1 complex is to mediate long-range chromatin interactions. Is LBD1 a member of the GATA2-TAL1-LMO4 complex and in this case, to what extent does LMO4 induction in EMPPs redirect 3 dimensional chromatin interactions?

2. Is there any evidence to suggest that the current findings shown in mice are relevant to human immune responses? Are there germline mutations associated with GATA2 deficiency syndrome that might interfere with LMO4 binding? Do human cells respond similarly to IL-33 exposure?

3. Do Gata2 G320D mice or LMO4-overexpressing mice exhibit altered responses to parasitic infections?

4. Figure 4: Please report the transduction frequency of bone marrow cells with LMO4 vector and the percent bone marrow repopulation with donor cells. What fraction of recipient GFP+ cells in BM are mCherry+? The readouts in this figure are

based on mCherry+ cells. Do the overall levels of various lineage cells change in these mice? If so, it may be possible to test the effects of LMO4 overexpression on parasite infection (see point 3, above).

Minor

1. Figure 1B- do peripheral blood counts of baso and eosinophils increase after infection?
2. Figure 2E: it appears that the PBS and IL-33 labeling of panels should be switched. Same for Figure 3D.
3. Typo at the bottom of page 4: "This lineage-instructive property of XXXX was further supported by induction of ectopic myeloid potential in preMegEs after Lmo4 overexpression (Fig.3f)."

Referee #2

(Remarks to the Author)

The manuscript presents a biologically interesting finding concerning the control of hematopoiesis by the alarmin IL-33 in the context of helminth infections. IL-33 is shown to promote the generation of basophils and eosinophils at the expense of erythrocytes via the induction of the transcription factor LMO4 in oligopotent EMPPs. Overall this aspect of the work is conceptually aligned with the previously established concept of emergency myelopoiesis.

Experimental concerns with this aspect involve the following:

1. Only gain of function experiments are performed with IL-33. Authors need to analyze the consequence of loss of IL-33 sensing via removal of the receptor ST2 in hematopoietic progenitors.
2. Can IL-33 signaling be shown to reprogram EMPPs towards eosinophil and basophil cell fates using a defined in vitro system? This would enable an analysis of the genes that are directly induced by IL-33 signaling including LMO4.

From a molecular standpoint the concept of redistribution of transcription factors on chromatin (in this case GATA2 via its differential interaction with LMO4) in the context of cell state or cell fate changes is of considerable interest but also has precedence.

Experimental issues to be addressed for this part include:

1. The interaction of GATA2 (WT) and G320D mutant proteins with LMO4, LMO2 and FOG-1 is analyzed by protein-protein interaction analyses exclusively. Complimentary analyses of protein-DNA interactions with cognate regulatory DNA motifs is needed.
2. Redistribution of GATA2 by LMO4 on chromatin is inferred from ATAC-seq experiments but not directly demonstrated using a variant of ChIP-seq (CUT and TAG).
3. The genetic experiment with the GATA2 G320D mutation does not involve phenotypic analysis with helminth infection or IL-33 administration, thus the overall experimental logic remains incomplete.

Referee #3

(Remarks to the Author)

In this manuscript titled "LMO4 is an IL-33-inducible master regulator of type 2 myelopoiesis that instructs myeloid fate via GATA factor chromatin re-allocation", Fagnan et al. report that helminth-infection-induced IL-33 drives upregulation of LMO4 expression in erythroid-primed multipotent progenitors (EMPPs) resulting in increased type 2 myeloid cell development from the progenitors of basophils, eosinophils and mast cells (BEMPs). The authors further show that LMO4 interacts with GATA2 in a complex that is distinct from the GATA2/LMO2/FOG1 complex and a GATA2 mutation that fails to interact with LMO4 has a deficiency in generating type 2 myeloid cells even when LMO4 is overexpressed. The fact that GATA proteins utilize distinct co-factors in different cell types and/or at different developmental stages to determine cell fates is not unexpected. The novelty of this study is the discovery of the IL-33-LMO4-GATA2 pathway in determining the differentiation of type 2 myeloid cells. However, many important conclusions are based on either bulk cell analyses or over-expression experiments. Therefore, the importance of this pathway under physiological conditions and the actual mechanism behind it cannot be determined.

Major concerns are:

1. A serious limitation of this study is lack of single cell analysis. Given progenitors are often very heterogenous even within a defined "pure" population, it is very difficult to judge whether the differences in gene expression in bulk reported in several conditions/treatments in this study are due to a change in gene expression per cell or due to a change in cell subsets that have different gene expression patterns within so-called "pure" progenitor population. Also, because of a rare population being analyzed, even a very small contamination of other cells can skew the results dramatically. In fact, two lymphoid cell-

associated genes *Gata3* and *Tcf7* were detected in some datasets but not in others. This limitation applies not only to RNA-Seq but also to ATAC-Seq.

2. Whether IL-33 has a direct effect on EMPPs is still questionable. IL-33 treatment of mice for several days can have several indirect effects. The authors should check IL-33 receptor expression at protein level on EMPPs. Can EMPPs directly respond to IL-33? More importantly, mixed chimeras using congenic WT together with *Il1r1* KO bone marrow should be analyzed with or without infection (or IL-33 injection). Furthermore, it is strange that IL-33 injection only induced eosinophils but not basophils (Figure 2d). Even for helminth infection when both eosinophils and basophils were increased, they were induced with a different kinetics (Figure 1b). Why? If a change in BEMPs is the chief mechanism, eosinophils and basophils should be affected similarly.

3. How about LMO4 expression at the protein level? Is it expressed by all EMPPs and BEMPs, or only by a subset among these cells. The degree of LMO4 induction by either infection or IL-33 is ~2 fold, (Figure 3a), whereas overexpression of LMO4 resulted in 4-5 fold higher than normal levels (Fig. S4d). Whether LMO4 is physiological important for this process should be tested in an LMO4 loss-of-function model.

4. While GATA2 G320 mutation specifically affects its interaction with LMO4 but not LMO2, it still could have other effects. The mutation may even have a major impact in earlier progenitors which leads to an abnormal EMPP population. In fact, the RNA-Seq results showed in Supplementary table 8 indicate that many important genes in determining cell fates including *Runx1*, *Runx2*, *IRF4*, *Spi1*, *Id2*, *Zbtb16*, *GATA1* and even *GATA2* itself have dramatically changed their expression in "EMPPs" from *GATA2* mutant mice. This makes data interpretation very difficult.

5. Because ATAC-Seq analyses were based on bulk population, it is difficult to determine whether GATA factor re-allocation causes changes in fate decision or alterations in cell population with different GATA occupancy is the reason.

Minor comments:

1. In several panels, in addition to reporting cell frequency, total cell counts should also be compared.

2. The columns in several supplementary tables seem to be mislabeled (i.e. $\log_2\text{FoldChange}$ in column B should be baseMean).

3. Many IFN-inducible genes were upregulated in EMPPs after *H. Polygyrus* infection (supplementary table 1). Any explanation?

4. *Il17ra*, *Il1r1* and *Crlf2* were only assessed at the mRNA level. How about their protein levels? Are they expressed by all EMPPs and BEMPs but at a different level, or just a subset of these cells. What are the effects of IL-25 or TSLP injection on EMPPs and BEMPs?

5. *Zfp1* downregulation and *Runx1* upregulation are accompanied by LMO4 expression. What are the functions of these transcription factors in LMO4-mediated cell fate determination?

6. *GATA2* haploinsufficiency has been reported. Any comments on why *Gata2*^{+/-} mice do not have any phenotype?

Version 1:

Reviewer comments:

Referee #2

(Remarks to the Author)

The authors have thoroughly addressed my major concerns with extensive new experiments, particularly in Figure 4.

In terms of the previous statement in the critique that "from a molecular standpoint the concept of redistribution of transcription factors on chromatin (in this case *GATA2* via its differential interaction with LMO4) in the context of cell state or cell fate changes is of considerable interest but also has precedence" the following references are provided as exemplars. The authors should include these references in their discussion:

Hosokawa H, et al., Transcription factor PU.1 represses and activates gene expression in early T cells by redirecting partner transcription factor binding. *Immunity* (2018), 49:782

Bender et al., Redistribution of PU.1 partner transcription factor RUNX1 binding secures cell survival during leukemogenesis *The EMBO Journal* (2024), 43:6291

Both of the above are in the context of the hematopoietic system but support for this idea exists in other developmental and cellular contexts. In the current manuscript the authors are able to delineate a TF:TF interaction surface that accounts for the competition and redistribution on chromatin with important consequences for cell fate determination.

Referee #3

(Remarks to the Author)

The authors have adequately addressed my previous concerns and the manuscript is significantly improved. No further comments.

General response:

We thank the referees and editor for their guidance and constructive comments – addressing the important points raised has, we believe, both strengthened our conclusions and added several new aspects to the manuscript.

In addition to the detailed responses to each point raised given below we will briefly outline the key additions made to the manuscript, in particular those related to points raised by more than one referee:

1. An important issue raised by referees 1 and 2 is whether the GATA-2–LMO4 interaction is important for the helminth immune response. To address this question, we have infected *Gata2* G320D mice and wild type littermates with *H. Polygyrus*, and observed that the mutant mice have a significantly higher parasite burden (new Fig. 6k,l), demonstrating that this is indeed the case.

Fig.6k,l: The number of worm settlement sites in the intestinal wall and worms in the intestine 14 days post *H. Polygyrus* infection. Note the significantly higher number of both in the *Gata2* G320D mutant genotype.

Note: Worm settlement sites are where the late-stage larvae mature – at 14 days the worms have vacated the sites (which appear as focal granulomas) and migrated to the lumen. The settlement sites are therefore at this stage essentially a historical record of the number of maturing larvae.

2. Another key point regards whether IL-33 acts directly on EMPPs to induce *Lmo4* (Referee 2, point 1 and Referee 3, point 2). This question is very pertinent, as IL-33 induces other cytokines (including IL-4, IL-5 and IL-13) in a variety of immune cell types (mast cells, ILC2s, Th2s), which could then induce *Lmo4* expression in EMPPs. Referee 3 suggested to generate bone marrow chimeras with cells from mice lacking the IL-33 receptor (ST2KO mice) to address this, an important consideration being that in bone marrow chimeras wild type versions of the IL-33 responsive cell types mentioned are present and can respond to IL-33, allowing for any indirect effects to be measured. We therefore generated 1:1 bone marrow-chimeras between ST2KO cells (or cells from their wild type controls; CD45.2 allotype) and wild type competitor cells (CD45.1 allotype), and treated these with IL-33 (new Fig 4a). In this setting fewer BEMPs were generated from ST2KO EMPPs than from wild type competitor EMPPs, whereas control CD45.2 cells showed no such difference. This difference was exacerbated upon IL-33 administration, showing that i) ST2/IL-33R mediates the IL-33 response of EMPPs, and ii) that ST2/IL-33R signalling also contributes to baseline BEMP formation (in the absence of exogenous IL-33 or infection). These conclusions were supported by analysis of the *in vitro* EMPP lineage potentials, where myeloid output was increased by IL-33 in wild type, but not ST2KO EMPPs after IL-33 challenge *in vivo* (new Fig 4a,b and Extended Data Fig 4a-d).

Fig 4a: Experimental workflow for generation and analysis of ST2KO bone marrow chimeras. After reconstitution with equal amounts of CD45.2 experimental and CD45.1 wild type helper bone marrow, recipient mice were injected daily with IL-33 and analysed after 2 and 6 days. **Fig.4b:** The reconstitution of CD45.1 and CD45.2 BEMPs in

control and IL-33–treated mice after 2 days is shown – note that decreased CD45.2/CD45.1 ratio in the ST2^{-/-}-transplanted recipients after IL-33, with a smaller, but significant decrease in PBS controls.

3. To address concerns regarding the heterogeneity of the populations studied (Referee 3, point 1) we have performed quantitative single cell transcriptome analysis of EMPPs, BEMPs and preMegEs in two key settings, namely *in vivo* IL-33–treatment of IL-33R/ST2 knockout chimeras and *Gata2* G320D mutant mice. This analysis clearly showed that EMPPs are a coherent population, distinct from both BEMPs and preMegEs (Fig 4c; Fig 6g), and that EMPPs express myeloid and Meg/Ery lineage-affiliated genes at levels intermediate between these two

committed progenitors (Fig 4d; Fig 6h). These perturbations (IL-33 treatment, G320D mutation) shifted the population mean, but did not significantly affect the population variance*, indicating that fate modulation occurs affecting the lineage bias of the overall EMPP population. In contrast, if lineage output was altered by selective expansion of an EMPPs subpopulation by IL-33, the range of signature expression would remain constant, but the distribution change, which was not observed. This is further supported by the distinct clusters formed by PBS and IL-33 treated WT EMPPs – in contrast, EMPPs lacking ST2 form a mixed cluster of PBS and IL-33-treated cells, consistent with these not being transcriptionally affected by IL-33 signalling. Importantly, *Lmo4* expression was directly regulated by IL-33 in EMPPs, as shown by the lack of IL-33 induction (and decreased baseline expression) of *Lmo4* expression in ST2^{-/-} EMPPs (Fig.4e).

Note: In the IL-33 experiment (panel 4c-e) there seems to be 2 preMegEs that were sorted into the control EMPP (WT PBS) population – this was not reproduced in subsequent experiments (with higher cell numbers) and is therefore likely a sorting error. For this reason, the F-test of the WT PBS and WT IL-33 EMPPs shows different variance – this is not seen if the two outliers are removed. No other EMPP combinations showed a significant F-test.

Fig. 4c: UMAP of the indicated cell populations generated from microfluidics-based gene expression analysis of myeloid (BEMP) and Meg-Ery (preMegE) lineage-affiliated genes. **Fig. 4d:** Distribution of ssGSEA scores of lineage programs in single cells. **Fig.4e:** Expression of *Lmo4* in the indicated cell populations.

Fig. 6g: UMAP generated as for 4c above. **Fig. 6h:** ssGSEA scores generated as in 4d above.

4. To address the relevance of our findings to humans (Referee 1, point 2) we have tested the response of human (h)EMPPs to IL-33. To address this point, we generated mice with human hematopoietic xenografts and used single cell expression profiling of the CD34+CD38+CD131+ progenitor subset (using the same approach as described in point 3) (new Fig 4f-h), which contains the human hEMPP, hBEMP and hMEP populations, to show that IL-33 administration led to *LMO4* up-regulation in hEMPPs (new Fig 4i) and expansion of the hBEMP population in vivo (new Fig 4j). Accordingly, in vitro colony forming assays of IL-33-treated CD131+ hCMPs showed an increase in myeloid-committed progenitors (Fig.4k), also confirming the direct role of IL-33 signalling. The IL-33–*LMO4* axis therefore acts analogously in mice and humans to promote type 2 myeloid lineage commitment.

Fig.4f: UMAP of clustering of human CD34+CD38–CD131+ progenitors generated from microfluidics-based gene expression analysis of myeloid (human basophil-mast cell progenitor - hBMCP) and Meg-Ery (human megakaryocyte-erythroid progenitor - hMEP) lineage-affiliated genes. Cells were isolated from xenografted NSGW41 mice treated with PBS, or human IL-33 (3 or 7 days) clustered together. **Fig.4g:** ssGSEA scores of the hMEP signature for each cluster from 4f. **Fig.4h:** ssGSEA scores of the hBMCP signature for each cluster from 4f. Clusters

were designated as BEMP (high hBMCP, low hMEP score), MEP (low hBMCP, high hMEP score) and EMPP (intermediate for both scores) **Fig.4i:** LMO4 expression in cells from PBS and IL-33–treated xenografts – note increased expression after 7 days of IL-33–treatment. **Fig.4j:** Distribution of PBS and IL-33–treated cells across clusters – note the increase in BEMPs after 7 days of IL-33 treatment. **Fig.4k:** Colony formation of human CD131+ CMPs cultured in the presence or absence of IL-33 – note the increase of myeloid-restricted colonies.

5. To understand the structural basis for the distinct functions of LMO2 and LMO4 we performed Alphafold modelling (in collaboration with Prof Erika Mancini, a structural biologist with specific expertise in LMO factors and their complexes). This showed that LMO2–LDB1–GATA-2 and LMO4–LDB1–GATA-2 complexes had distinct conformations: GATA-2 contacts the LMO4 LIM-2 domain in a manner distinct from that of the LMO2 LIM-2 domain, leading GATA-2 to rotate relative to its position in the LMO2 complex. This causes GATA-2 valine (V)296, the key FOG-1 contact residue (PMID: 21844373), to move to an internal position within the LMO4–LDB1–GATA-2 complex, preventing FOG-1 from binding (Fig 5d,e). To validate the modelling, we replaced the homologous part of LMO2 LIM-2 domain with the LMO4 sequence (Fig 5f). This was sufficient to i) cause V296 internalisation (Fig 5g); ii) prevent FOG-1 interaction with the LMO2–LDB1–GATA-2 complex (Fig 5h) and iii) confer myeloid-instructive activity on LMO2 (Fig 5i). Overall, this analysis provides a structural basis for the ability of LMO4 to displace FOG-1 from GATA-2 and strengthens the correlation between FOG-1 displacement and myeloid lineage choice.

Fig.5d: AlphaFold3 generated model of LMO2/LDB1/GATA-2 ZnF1 – FOG-1 was docked to the predicted structure based on the NMR structure from Wilkinson-White et al. (PMID: 21844373). V296, the key FOG-1

contact residue in GATA-2 ZnF1, is shown. **Fig.5e**: AlphaFold3 generated model of LMO4/LDB1/GATA-2 ZnF1. Note that Valine 296 is now internal to the structure, and FOG-1 docking not possible. **Fig.5f**: Domain structure of LMO2, LMO4 and chimeras (C1, C2) where LMO2 domains were substituted with the equivalent LMO4 sequence. **Fig.5g-i**: The LMO-C2 chimera has internal V296 position (g), instructs myeloid fate (h) and excludes FOG-1 from its GATA-2 complex (i). The LMO4 LIM2 domain is therefore sufficient to confer LMO4-like activity on LMO2.

Note: LMO2 overexpression has significant growth-inhibitory effect, and relatively few colonies were obtained from LMO2 and LMO2-C1 transduced EMPPs – these were, however, essentially all Meg-Ery.

6. To substantiate the GATA factor re-allocation inferred from ATAC-seq-based foot-printing, we have performed ChIP-seq using the HPC-7 hematopoietic progenitor cell line, which has been used extensively for the study of TF colocalization. Expression of LMO4 in HPC-7 cells caused significant GATA-2 re-allocation: >40% of the detected GATA-2 binding sites were either lost or gained (Fig.4o). Importantly, the GATA-2 sites lost were highly enriched in Meg-Ery-specific GATA binding sites, whereas those gained were highly enriched in myeloid-specific GATA binding sites (Fig.4p). Therefore, LMO4 induces large-scale re-allocation of GATA-2 from Meg-Ery to myeloid chromatin.

Fig.5o: Heatmap of GATA-2 ChIP signal at HPC-7 GATA-2 peaks unaffected (left), increased (center) and depleted (right) by LMO4 expression. **Fig.5p**: AlphaFold3 generated model of LMO4/LDB1/GATA-2 ZnF1. Note that Valine 296 is now internal to the structure, and FOG-1 docking not possible. **Fig.5f**: Domain structure of LMO2, LMO4 and chimeras (C1, C2) where LMO2 domains were substituted with the equivalent LMO4 sequence. **Fig.5g-i**: The LMO-C2 chimera has internal V296 position (g), instructs myeloid fate (h) and excludes FOG-1 from its GATA-2 complex (i).

Together, these additional experiments and analyses confirm the key role of Lmo4 in both baseline and emergency type 2 myelopoiesis, the latter in response to IL-33. Furthermore, we now show that IL-33 directly induces Lmo4 in EMPPs in both mice and humans, provide a structural basis for the ability of LMO4 to displace FOG-1 from GATA-2, and show this leads to GATA factor re-allocation to myeloid-specific chromatin using both in silico foot-printing and ChIP-seq. We now provide two distinct loss-of-function models (GATA-2 mutation that blocks LMO4 interaction, Lmo4 knockdown) that both show a critical role of Lmo4 in myeloid lineage commitment. Together with the structural analysis shown above, this provides a comprehensive mechanistic basis for how parasite infection initiates type 2 myelopoiesis, and demonstrates its physiological relevance and evolutionary conservation. We believe these results fill a considerable knowledge gap regarding how type 2 myelopoiesis is initiated, and in the process identify a master regulator of these lineages, as well as previously uncharacterized molecular mechanisms that are likely to be of general importance.

The detailed responses to the individual points made by the referees are found below.

Referees' comments:

Referee #1 (Remarks to the Author):

After exposure to parasitic infection, hosts demonstrate a type 2 innate myeloid immune response with increased production of basophils, eosinophils and mast cells (BEM) through incompletely understood mechanisms. The authors study this response in mice infected with the roundworm *H. polygyrus*. It was previously established that parasite infection causes erythroid-primed multipotent progenitors (EMPPs) to differentiate into BEM progenitors at the expense of pre-megakaryocyte-erythroid progenitors (preMegEs), resulting in enhanced production of BEM and reduced red blood cell and platelet production. The current study shows that this lineage fate switch is caused by parasite-induced release of IL-33, which causes EMPPs to upregulate LMO4, a member of an activating transcription factor complex that also contains GATA1 or GATA2, FOG1 and TAL1 (and LDB1). In meg and erythroid lineages, this complex usually contains LMO2. However, during parasitic infection, IL33 caused the replacement of LMO2 with LMO4, which displaced FOG1 and redirected the complex from Meg-Erythroid genes to BEM-related ones, induced expression of the latter and caused a corresponding shift in the lineage output. The authors note this to be an interesting example of transcription factor re-allocation as a mechanism of lineage commitment.

This paper characterizes an interesting mechanism for lineage specification in response to a clinically relevant physiological stress. The lineages examined are relatively rare, difficult to study and poorly understood. The experiments are technically sound and the paper is clearly written. Suggestions for improvement include:

Major

1. Additional mechanistic insights into the key observations would strengthen the paper. Additional outstanding questions include:

a. How do differences in the structures of LMO2 vs LMO4 modulate distinct interactions with FOG1 and GATA2 G320D (or lack thereof)? It should be possible to investigate this by structural modeling combined with additional mutagenesis.

Response:

*We used AlphaFold3 to model the complexes between LMO2–LDB1–GATA-2 and LMO4–LDB1–GATA-2, using the published NMR structure of as the starting point (i.e. using the same protein sequences, as these were known to be sufficient for complex formation). This showed that LMO2 and LMO4 contacted GATA-2 in different ways, leading to distinct complex conformations. Importantly, while in the LMO2–LDB1–GATA-2 complex the critical FOG-1 contact residue, valine (V)296, was external to the structure and allowed docking of FOG-1 (Fig.5d), in the LMO4–LDB1–GATA-2 complex this residue was internal and inaccessible (Fig.5e). Substituting the LIM-2 domain of LMO2 with that of LMO4 (Fig.5f) was sufficient to generate a V296-inaccessible conformation (Fig.5g). This chimera also excluded FOG-1 from its complex with GATA-2 (Fig.5i), and had myeloid-instructive activity in EMPPs (Fig.5h), providing further linkage of these activities. These observations provide a straightforward molecular basis for how LMO4 displaces FOG-1. Interestingly, we find that expression of *Zfpm1* (which encodes FOG-1) in EMPPs completely blocks myeloid differentiation (see response to Referee 3, point 5), indicating that the LMO4–FOG-1 competition can work in both directions.*

b. What is the mechanism by which IL-33 induces LMO4?

Response:

This is a very relevant point – as the role of LMO4 in myeloid lineage commitment has not been previously shown, and according little attention has been paid to how it is regulated in haematopoiesis. To address this point, we have examined the key signalling pathways emanating from the IL-33 receptor (ERK/MAPK, NF-κB, p38). Of these, only inhibition of NF-κB signalling (with IKK16) prevented LMO4 up-regulation in primary EMPPs after IL-33 treatment. These data have been added (new Extended Data Fig. 4e).

c. A major function of the GATA-TAL-LMO2-LDB1 complex is to mediate long-range chromatin interactions. Is LDB1 a member of the GATA2-TAL1-LMO4 complex and in this case, to what extent does LMO4 induction in EMPPs redirect 3-dimensional chromatin interactions?

Response:

The effect of LMO4 is to re-distribute a considerable proportion of GATA-2, as inferred from ATAC-seq analysis and now shown directly by ChIP-seq – this almost certainly lead to chromatin reorganisation.

It is, however, difficult to address this point directly in primary cells, as it will require mapping the 3D chromatin structure of EMPPs, and global chromatin conformation analysis (such as Hi-C) requires millions of cells, several orders of magnitude more than the number of EMPPs that can be readily isolated (100s-1000s of cells/mouse), an issue that is further exacerbated for LMO4 perturbation analysis where only a subset of cells would express (or lack) LMO4. Some recent protocols have reduced the cell number needed to around 100,000 (e.g. Easy-Hi-C: PMID: 36427146) at cost of lower efficiency, but this is still far more than can realistically be obtained, especially if LMO4 overexpression is required in EMPPs in vivo.

Therefore, we have not addressed this point experimentally, but added this consideration to the discussion.

2. Is there any evidence to suggest that the current findings shown in mice are relevant to human immune responses? Are there germline mutations associated with GATA2 deficiency syndrome that might interfere with LMO4 binding? Do human cells respond similarly to IL-33 exposure?

Response:

GATA-2 mutations found in GATA-2 deficiency syndrome affect (or delete) ZnF2, or remove critical regulatory elements that control GATA2 expression (see e.g. Wlodarski 2017; PMID: 28637621). Mutations in ZnF1, where LMO4 binds, have not been reported. LMO4 is therefore unlikely to play a role in this syndrome.

To explore whether IL-33 instructs myeloid commitment in the human system we performed both in vivo and in vitro experiments. First, we established human hematopoiesis in xenografted NSGW41 mice, followed by treatment with IL-33 for 3 or 7 days, or vehicle treatment. We next isolated the CD34+CD38+CD131+ progenitor cells (this population contains the GATA1-expressing myeloid and erythroid/megakaryocytic progenitors in humans – see Drissen et al., 2019 – cited) from the xenografted mice and performed microfluidics-based single cell profiling using Meg-Ery (MEP) and Bas-Mas progenitor (BMCP) signatures (from PMID: 38514887). We identified the human BEMP, EMPP and preMegE equivalent clusters from their signature expression (new Fig. 4f-h), and observed that IL-33 treatment up-regulated LMO4 in EMPPs (new Fig. 4i) and led to expansion of BEMPs (Fig. 4j). Finally, we treated purified human CD131+ CMPs with recombinant IL-33, followed by colony forming assays, observing an increase in myeloid (i.e. basophil/eosinophil) colony formation (Fig. 4k). Therefore, IL-33 regulation of LMO4 in EMPPs and subsequent myeloid lineage commitment is conserved between murine and human hematopoiesis.

3. Do Gata2 G320D mice or LMO4-overexpressing mice exhibit altered responses to parasitic infections?

4. Figure 4: Please report the transduction frequency of bone marrow cells with LMO4 vector and the percent bone marrow repopulation with donor cells. What fraction of recipient GFP+ cells in BM are mCherry+? The readouts in this figure are based on mCherry+ cells. Do the overall levels of various lineage cells change in these mice? If so, it may be possible to test the effects of LMO4 overexpression on parasite infection (see point 3, above).

Response (points 3+4):

The proportion of cells expressing LMO4 in the lentivirally transduced bone marrow chimeras is 2-4% in peripheral blood and 4-9% in the bone marrow (as can be seen in Fig.3i and Extended Data Fig.3d-g). This limits any physiological impact of these cells in this model.

We therefore tested how the Gata2 G320D mutation affects parasite control after H. Polygyrus infection (as also suggested by Referee 2). We observed that after infection with H. Polygyrus the G320D mice exhibited a notable increase in parasite burden, with significantly higher number of worms and worm settlement sites detected in the intestine, indicating loss of control of the parasite infection (Fig. 6i,j). This was accompanied by an attenuated anti-parasite immune response, with significantly lower spleen weight and decreased basophil and eosinophil numbers in both bone marrow and spleen.

As already seen in other experiments there was skewing of progenitors with worm-infected G320D mice having decreased BEMPs and increased preMegEs in the bone marrow. These observations are all consistent with decreased eosinophil (and basophil) production in response to parasite infection leading to impaired anti-parasite immunity, and with this defect being initiated by decreased commitment to a myeloid fate of EMPPs.

Minor

1. Figure 1B - do peripheral blood counts of baso and eosinophils increase after infection?

The increase of eosinophils and basophils induced by *H. Polygyrus* infection has previously been reported in peripheral blood (PMID: 25404305 and PMID: 22930820). We measured the abundance of these cell types in bone marrow and spleen only. The number of eosinophils and basophils found in spleen of naïve and infected mice is shown below. As this increase is well-characterized, we did not include the data in the manuscript.

Fig.R1. The total number of splenic eosinophils (left) and basophils (right) as determined by flow cytometry of naïve and *H. Polygyrus*-infected mice 11 days post-infection.

2. Figure 2E: it appears that the PBS and IL-33 labeling of panels should be switched. Same for Figure 3D.

These panels are correctly labelled – note the higher number of mast cell colonies and reduced Meg-Ery colonies in the IL-33 condition.

3. Typo at the bottom of page 4: “This lineage-instructive property of XXXX was further supported by induction of ectopic myeloid potential in preMegEs after Lmo4 overexpression (Fig.3f).”

We thank the referee for spotting this omission – LMO4 has been added to this sentence.

Referee #2 (Remarks to the Author):

The manuscript presents a biologically interesting finding concerning the control of hematopoiesis by the alarmin IL-33 in the context of helminth infections. IL-33 is shown to promote the generation of basophils and eosinophils at the expense of erythrocytes via the induction of the transcription factor LMO4 in oligopotent EMPPs. Overall this aspect of the work is conceptually aligned with the previously established concept of emergency myelopoiesis.

General comments:

We thank the referee for their positive comments. Indeed, one of the key reasons for undertaking this work was that while emergency myelopoiesis is well studied for the neutrophil/monocyte lineages, far less is known about the mechanisms underlying increased type 2 myeloid cell output in response to infection, a significant knowledge gap that this manuscript aims to close.

Experimental concerns with this aspect involve the following:

1. Only gain of function experiments are performed with IL-33. Authors need to analyze the consequence of loss of IL-33 sensing via removal of the receptor ST2 in hematopoietic progenitors.

2. Can IL-33 signaling be shown to reprogram EMPPs towards eosinophil and basophil cell fates using a defined in vitro system? This would enable an analysis of the genes that are directly induced by IL-33 signaling including LMO4.

Response (points 1+2):

As suggested, we have analysed the effect of Il1r1/ST2 loss on progenitor differentiation in response to IL-33. For this purpose, we generated bone marrow chimeras by co-transplanting CD45.2 ST2 KO (or wild type control) cells, along with CD45.1 wild type competitor bone marrow cells, into lethally irradiated CD45.1 recipients. After reconstitution mice were injected with IL-33 or vehicle (PBS), and the progenitor compartment examined after 2 and 6 days. At both timepoints we observed a small, but significant, decrease in the proportion of ST2 KO BEMPs relative to wild type BEMPs in vehicle injected mice, and a far greater difference upon IL-33 exposure. IL-33 signalling via ST2 therefore promotes myelopoiesis at the level of BEMP formation also at steady state, with a

further increase in BEMPs upon IL-33 administration). These data have now been included (Fig.4a,b, Extended Data Fig.4a-d).

To correlate the observed phenotypes to *Lmo4* regulation we performed microfluidics-based qRT-PCR on single wild type and ST2 KO EMPPs isolated from both control and IL-33-treated recipients. This method was chosen (rather than single cell RNAseq) due to its high sensitivity, which allows accurate quantification of mRNA levels at the single cell level (as opposed to RNAseq-based readouts which are largely digital in nature due to technical dropout). This showed the expected increase in *Lmo4* expression in IL-33-treated wild type EMPPs compared to control (PBS treated) EMPPs, an increase that was not observed in IL-33 exposed ST2 KO EMPPs (Fig.4e). Furthermore, we could show that baseline *Lmo4* expression was decreased in ST2KO EMPPs compared to wild type EMPPs in the absence of IL-33 administration. Considering the small, but significant, decrease in baseline BEMP formation from ST2KO EMPPs this indicates that the IL-33-LMO4 axis regulates commitment to type II myelopoiesis at both steady state and in response to IL-33/infection. We have added the data (Fig.4c-e, Extended Data Fig.4a-d) and a discussion of these findings.

These experiments clearly showed that IL-33 acts directly on EMPPs to induce transcriptional reprogramming and myeloid bias. As the experiments with ST2 chimeras allowed us to measure effects on both cell fate and gene expression on EMPPs directly when induced by IL-33 in vivo we used in vitro analysis to i) confirm that *Lmo4* up-regulation is a direct consequence of IL-33 expression, and ii) use this assay to explore the signalling pathways controlling IL-33-mediated *Lmo4* up-regulation – this analysis showed that NF- κ B signalling was the key *Lmo4* inducer (Extended Data Fig.4e).

In the absence of ST2 we observe little, if any, transcriptional or functional effect of IL-33 on EMPPs, indicating that the bulk of the IL-33 response is due to direct signalling. However, we cannot exclude other indirect mechanisms by which IL-33 could act on the progenitor compartment via release of other cytokines, and we have included this consideration in the discussion of the data.

From a molecular standpoint the concept of redistribution of transcription factors on chromatin (in this case GATA2 via its differential interaction with LMO4) in the context of cell state or cell fate changes is of considerable interest but also has precedence.

Response:

We are not entirely sure which precedent(s) the referee is referring to – as far as we are aware this is the first time that a lineage-instructive factor has been shown to drive a lineage decision via re-allocation of a transcription factor between lineage-specific chromatin domains. As discussed in the response to point 2 below, we have now substantiated this finding showing GATA-2 protein reallocation by LMO4 in the HPC-7 progenitor cell line using ChIP-seq.

Experimental issues to be addressed for this part include:

1. The interaction of GATA2 (WT) and G320D mutant proteins with LMO4, LMO2 and FOG-1 is analyzed by protein-protein interaction analyses exclusively. Complimentary analyses of protein-DNA interactions with cognate regulatory DNA motifs is needed.

Response:

To address the effect of the G320D mutation in a chromatin context, and in particular whether this mutation selectively affects LMO4 interaction with GATA-2, we have performed ChIP-seq from 293T cells co-transfected with GATA-2 (wild type or G320D mutant), LMO2 and LMO4. This showed that the G320D mutation caused a selective decrease in LMO4 binding to GATA-2 bound loci, with no effect on LMO2 (Fig 5h), consistent with the co-IP data.

2. Redistribution of GATA2 by LMO4 on chromatin is inferred from ATAC-seq experiments but not directly demonstrated using a variant of ChIP-seq (CUT and TAG).

Response:

As the low number of EMPPs in the bone marrow (and even more so LMO4-transduced EMPPs) makes ChIP-seq and related assays highly challenging using primary cells, we sought to identify a cell line suitable for these experiments. HPC-7 cells are an immortalized multipotent hematopoietic progenitor cell line (PMID: 11495708) that has been widely used for analysing TF occupancy in hematopoiesis, including GATA-2 (PMID 20887958, 26809507, 25319994, 25319996, 26901438), and we believe it represents the best possible system for this analysis. HPC-7 cells express *Gata2*, *Lmo2* and *Zfp1* (encoding FOG-1), but not LMO4, making it suitable for

studying the effect of LMO4 expression, and LMO4 vs. LMO2/FOG-1 competition in particular, on GATA-2 chromatin distribution. By performing GATA-2 ChIP-seq on control and LMO4-transduced HPC7 cells, we observed very significant GATA-2 redistribution by LMO4: of 12947 GATA-2 binding sites, 5343 were lost, with 1107 new sites gained. 7604 sites were occupied both in the presence and absence of LMO4 (new Figure 4m). We next generated BEMP- and preMegE-specific GATA-2 binding signatures using TOBIAS. GSEA analysis using these signatures showed that LMO4 suppressed GATA-2 binding to preMegE-specific sites and enhanced binding to BEMP-specific sites (new Figure 4n). Together, these observations show that LMO4 is capable of redistributing GATA-2, and that this involves depleting GATA-2 from Meg-Ery-specific chromatin and relocating it to myeloid-specific chromatin. Therefore, direct measurement of the effects of LMO4 on GATA-2 chromatin distribution confirmed the conclusion from ATAC-based inference of GATA factor binding.

3. The genetic experiment with the GATA2 G320D mutation does not involve phenotypic analysis with helminth infection or IL-33 administration, thus the overall experimental logic remains incomplete.

Response:

We have addressed this in our general response and in the response to referee 1, point 4, showing that mice carrying the G320D mutation are indeed defective in their ability to control parasite infection.

Referee #3 (Remarks to the Author):

In this manuscript titled “LMO4 is an IL-33-inducible master regulator of type 2 myelopoiesis that instructs myeloid fate via GATA factor chromatin re-allocation”, Fagnan et al. report that helminth-infection-induced IL-33 drives upregulation of LMO4 expression in erythroid-primed multipotent progenitors (EMPPs) resulting in increased type 2 myeloid cell development from the progenitors of basophils, eosinophils and mast cells (BEMPs). The authors further show that LMO4 interacts with GATA2 in a complex that is distinct from the GATA2/LMO2/FOG1 complex and a GATA2 mutation that fails to interact with LMO4 has a deficiency in generating type 2 myeloid cells even when LMO4 is overexpressed. The fact that GATA proteins utilize distinct co-factors in different cell types and/or at different developmental stages to determine cell fates is not unexpected. The novelty of this study is the discovery of the IL-33-LMO4-GATA2 pathway in determining the differentiation of type 2 myeloid cells. However, many important conclusions are based on either bulk cell analyses or over-expression experiments. Therefore, the importance of this pathway under physiological conditions and the actual mechanism behind it cannot be determined.

General comments:

We thank the referee for their insightful comments. We agree that an important aspect of this manuscript is that it addresses the considerable knowledge gap regarding how type 2 myeloid cells are specified and how emergency type 2 myelopoiesis is initiated. The referee is also correct that GATA cofactor use differs between cell types – this has been clear since the seminal work by the Orkin lab showing the selective importance of FOG-1 in megakaryocyte and erythroid lineages. However, we still have limited knowledge of the interplay between these co-factors, how it is regulated, and the developmental and physiological consequences thereof. This is a question of significant relevance, given the pervasive importance of GATA factors in development and disease. We here contribute to this by identifying LMO4 as a previously unrecognized master regulator of both baseline and emergency type 2 myelopoiesis, identifying IL-33 as a inducer of its expression in EMPPs, the critical oligopotent progenitor where commitment to a type 2 myeloid fate occurs, and showing that LMO4 acts by displacing FOG-1 from GATA factors (and GATA-2 in particular) and relocating these to myeloid-specific genes. We now add additional mechanistic evidence, demonstrating the requirement of ST2 on EMPPs for IL-33 to induce molecular and functional lineage bias, and providing a structural basis for the ability of LMO4 to displace FOG-1 from GATA-2.

The referee is of course right that overexpression analysis needs to be treated cautiously, as artefactual results may occur. We would like to point out that throughout this study Lmo4 overexpression analyses were accompanied by corresponding loss-of-function analyses (G320D mutant), and the converse phenotype was consistently observed. We now add to this by using another loss-of-function model (Lmo4 knockdown) to confirm the role of LMO4 in myeloid commitment, both baseline and IL-33 induced. Our view is that combining gain- and loss-of-function studies provides the most comprehensive analysis possible. We would also like to point out that Lmo4 gain-of-function analysis is essential to ascertain whether the up-regulation of Lmo4 (as seen upon IL-33 exposure) is sufficient to generate myeloid bias of EMPPs. In summary, we do not consider the use of Lmo4 overexpression to be a weakness of the study, but a strength, and it both complements the loss-of-function analysis and mimics the physiological effect of IL-33 and parasite infection.

The other point raised relates to the use of bulk (as opposed to single cell) analysis. As avid users of single cell analysis (both functional and molecular) we fully appreciate the advantages of this type of technology, in particular when deconvoluting heterogeneity of complex populations. However, once this has been achieved, and markers identified that can isolate functionally and molecularly homogeneous cell populations, bulk analysis has some key advantages: first, it allows more powerful bioinformatics to be applied (in this case TOBIAS, which does not work at the single cell level) and secondly, it allows direct correlation between molecular and functional changes, as the same population can be used in parallel for molecular profiling and functional studies. In contrast, populations identified by e.g. clustering of single cell data cannot be readily isolated for functional analysis. Of course, the use of bulk analysis has as a prerequisite that the populations used are functionally and molecularly homogeneous. In the present case, the BEMP and EMPP populations were identified and characterized by ourselves (Drissen et al., 2016 – cited) based on extensive single cell analysis, both functional and molecular, as were the molecular markers that were used to isolate them, and shown to be molecularly coherent and functionally distinct (preMegEs were described previously (Pronk et al, 2007 – cited)). The additional single cell profiling now performed fully support these previous studies, and in particular shows that EMPPs, while responding to perturbations with altered molecular lineage bias, remain a coherent population distinct from both preMegEs and BEMPs. Based on this we believe the use of bulk analysis is fully justified and appropriate.

Major concerns are:

1. A serious limitation of this study is lack of single cell analysis. Given progenitors are often very heterogenous even within a defined “pure” population, it is very difficult to judge whether the differences in gene expression in bulk reported in several conditions/treatments in this study are due to a change in gene expression per cell or due to a change in cell subsets that have different gene expression patterns within so-called “pure” progenitor population. Also, because of a rare population being analyzed, even a very small contamination of other cells can skew the results dramatically. In fact, two lymphoid cell-associated genes Gata3 and Tcf7 were detected in some datasets but not in others. This limitation applies not only to RNA-Seq but also to ATAC-Seq.

Response:

The referee raises an important point, which is that if a cell population is composed of functionally distinct sub-populations, changes to the differentiation bias of the population may be due to altered proportions of the constituent subpopulations, rather than a change to the differentiation bias of individual cells. A classic example of this is the phenotypic common myeloid progenitor (CMP), which was subsequently shown to contain separate Meg-Ery and neutrophil-monocyte progenitor populations, rather than oligo-potent progenitors containing all of these potentials. This consideration is, as pointed out, central to understanding how a perturbation (TF mutation, cytokine stimulation) changes the differentiation bias of a cell population: if the population is heterogeneous with each sub-population highly biased towards one differentiation outcome, perturbations would change the proportions of the subpopulations, and expression signatures of the output lineages would be expected to show a biphasic distribution, as such lineage priming generally correlates with differentiation outcome. In contrast, for a functionally homogeneous population a perturbation that alters differentiation bias would be expected to cause a shift in the mean signature expression.

As discussed above, we have previously used extensive single cell profiling to identify BEMPs and EMPPs as homogeneous progenitor populations. However, to further address this point, we have applied single cell transcriptome analysis in two key settings, namely to EMPPs from the in vivo administration of IL-33 to ST2KO (and control) bone marrow chimeras (as discussed in the response to Referee 2, point 1), and to EMPPs from Gata2 G320D and wild type control mice. To directly address the question of whether the EMPP population contains distinct cellular subsets that are biased towards Meg-Ery or myeloid differentiation, respectively, we identified genes selectively up-regulated in BEMPs and preMegEs, and analysed the expression of these lineage signatures in single EMPPs from the different experimental conditions described above, as well as preMegEs and BEMPs as reference populations. For expression analysis we used microfluidics-based real time qRT-PCR (Fluidigm Biomark platform), which allows for quantification of mRNA expression at the single cell level (as opposed to the essentially digital readout obtained from scRNAseq).

First of all, clustering of cells using the microfluidics-based gene expression data showed that EMPPs formed a coherent population, clearly distinct from BEMPs and preMegEs (Fig.4c and Fig.6g). Importantly, this remained the case for all perturbations – while IL-33-treated EMPPs formed a sub-cluster distinct from control EMPPs (Fig.4c), they remained within the larger EMPP cluster. ST2KO EMPPs similarly formed a distinct EMPP sub-cluster – in this case IL-33 treated cells did not induce any distinct clustering, consistent with these being refractory to IL-33 exposure. Importantly, this strongly supports that IL-33 directly reprograms EMPPs (as opposed to acting through an intermediary cytokine). Similarly, wild type and Gata2 G320D mutant EMPPs formed a single cluster, distinct from BEMPs and preMegEs (Fig.6g).

Furthermore, from these data the expression levels of BEMP- and preMegE-specific signature genes could be accurately quantified and used to generate signature scores for each cell generated using ssGSEA. We found that in wild type EMPPs these signatures are expressed at a level intermediate between preMegEs and BEMPs (new Fig.4d and Fig.6h), as expected. The distribution of the EMPP ssGSEA scores did not reveal any distinct subpopulations. The mean of the distribution was shifted towards the BEMP population by IL-33 treatment, and towards the preMegE population by ST2 knockout, in line with the observed shift in balance between myeloid and Meg-Ery differentiation. Also, in ST2KO EMPPs signature expression was not significantly altered by IL-33 exposure, consistent with their differentiation being unaffected by IL-33 in both in vivo and in vitro progenitor readout. Finally, there was a significant down-regulation of the BEMP signature, and conversely up-regulation of the preMegE signature, in G320D EMPPs compared to control wild-type EMPPs. However, in all cases the EMPP population remains coherent and distinct from BEMPs and preMegEs, consistent with the observed clustering.

Regarding the expression of lymphoid genes, these are, as mentioned, seen stochastically (i.e. in some, but not all, samples), and likely represent low levels of contamination during cell sorting. As such, they are unlikely to contribute to statistically significant differences, as the variance in their abundance will be high. Furthermore, such random contamination would be equally present in control and experimental samples, and would thus cancel each other out in subsequent molecular analyses.

2. Whether IL-33 has a direct effect on EMPPs is still questionable. IL-33 treatment of mice for several days can have several indirect effects. The authors should check IL-33 receptor expression at protein level on EMPPs. Can EMPPs directly respond to IL-33? More importantly, mixed chimeras using congenic WT together with Il1r1 KO bone marrow should be analyzed with or without infection (or IL-33 injection). Furthermore, it is strange that IL-33 injection only induced eosinophils but not basophils (Figure 2d). Even for helminth infection when both eosinophils and basophils were increased, they were induced with a different kinetics (Figure 1b). Why? If a change in BEMPs is the chief mechanism, eosinophils and basophils should be affected similarly.

Response:

The point regarding whether IL-33 acts directly on EMPPs to promote BEMP differentiation is well taken. As discussed above (referee 2, point 1) we have generated chimeric mice with Il1r1/ST2 KO cells, and treated them with IL-33. This clearly showed that there is an intrinsic defect in BEMP formation from EMPPs in response to IL-33. Clearly, this does not preclude that IL-33 promotes myeloid cell development by other mechanisms (such as its well-established ability to increase production of IL-5), and we have included this consideration in the discussion. However, from this analysis, and from the accompanying gene expression analysis discussed in the response to point 1, it seems clear that direct signalling through the IL-33 receptor is the main contributor to transcriptional EMPP reprogramming and increased BEMP formation.

Regarding the differences in basophil and eosinophil output this is not necessarily surprising – production of mature lineages is regulated at several points downstream of BEMPs, including by IL-5 (which selectively promotes eosinophil production). Furthermore, the kinetics of myeloid cell differentiation from BEMPs differs between cell types (see e.g. Drissen et al., 2016 - cited), with eosinophil production taking longer than formation of mast cells (which are molecularly similar to basophils), which may affect readouts. It is therefore not surprising that differences between these lineages is observed. Importantly, in steady state (e.g. LMO4 overexpression (Fig.3k), G320D mutation (Fig.6a)) effects are similar for basophils and eosinophils.

3. How about LMO4 expression at the protein level? Is it expressed by all EMPPs and BEMPs, or only by a subset among these cells. The degree of LMO4 induction by either infection or IL-33 is ~2 fold, (Figure 3a), whereas overexpression of LMO4 resulted in 4-5 fold higher than normal levels (Fig. S4d). Whether LMO4 is physiologically important for this process should be tested in an LMO4 loss-of-function model.

Response:

We have now examined the LMO4 protein level in EMPPs, BEMPs and preMegEs using intracellular flow cytometry (Fig.3g). As expected, BEMPs showed the highest and preMegEs the lowest levels of LMO4 protein, with EMPPs intermediate between the two. Importantly, also in this analysis there was no evidence for subpopulations with distinct LMO4 expression in any of the progenitor populations.

Regarding a loss-of-function model, this is already provided by the Gata2 G320D mutation. In addition to this, we now use in vivo Lmo4 knockdown to show that Lmo4 is required for both baseline and IL-33-induced BEMP formation from EMPPs (Extended Data Fig.8a-d). The two loss-of-function models therefore have the same effect, providing cross-validation of the results.

4. While GATA2 G320 mutation specifically affects its interaction with LMO4 but not LMO2, it still could have other effects. The mutation may even have a major impact in earlier progenitors which leads to an abnormal EMPP population. In fact, the RNA-Seq results showed in Supplementary table 8 indicate that many important genes in determining cell fates including Runx1, Runx2, IRF4, Spi1, Id2, Zbtb16, GATA1 and even GATA2 itself have dramatically changed their expression in “EMPPs” from GATA2 mutant mice. This makes data interpretation very difficult.

5. Because ATAC-Seq analyses were based on bulk population, it is difficult to determine whether GATA factor re-allocation causes changes in fate decision or alterations in cell population with different GATA occupancy is the reason.

Response (points 4+5):

Given that the G320D EMPP population has altered lineage bias (i.e. decreased BEMP and increased preMegE output), it is not surprising that TF gene expression is altered – in fact, it would be expected. Indeed, the single cell transcriptome analysis now provided clearly shows that EMPPs alter their molecular lineage bias in response to perturbations, in a manner reflecting their altered functional lineage bias. However, as discussed above, it is equally clear that they remain a coherent population, molecularly distinct from the more restricted preMegE and BEMP progenitors. This analysis also shows that both IL-33 exposure and G320D mutation shift the molecular lineage bias by altering the population mean, but not the variance, and there is no evidence that either involves altering the prevalence of EMPP sub-populations, for which we can find no evidence.

While the consistently opposing phenotypes of G320D mutation and LMO4 overexpression clearly indicates that loss of LMO4 interaction is the key effect of the mutation the referee’s point that other GATA-2 functions could be affected is well taken. We have now, as discussed in the response to point 3, performed Lmo4 knockdown, which shows the same effect as G320D (i.e. loss of myeloid commitment in EMPPs), further substantiating that the effect of the mutation is to block LMO4 interaction.

Also, for clarification: Gata1, Gata2, Spi1, Irf4, Runx2 and Zbtb16 are not differentially expressed in the analysis in Sup Table 8 – of the genes mentioned only Runx1 (Padj=0.04) and Id2 (Padj=2.6e-10) show a significant difference. As shown below, Runx1 does not seem to have significant lineage instructive capacity in EMPPs, and we are not aware of any studies showing a role for Id2 in this setting.

Minor comments:

1. In several panels, in addition to reporting cell frequency, total cell counts should also be compared.

Response:

This has been added.

2. The columns in several supplementary tables seem to be mislabeled (i.e. log2FoldChange in column B should be baseMean).

Response:

We thank the referee for spotting this - it has been corrected.

3. Many IFN-inducible genes were upregulated in EMPPs after H. Polygyrus infection (supplementary table 1). Any explanation?

Response:

It is well-established that H.Polygyrus infection induces both type I (e.g. PMID: 28196762) and type II interferon (e.g. PMID: 34671159) responses. Consistent with this the GSEA analysis shows up-regulation of both interferon-alpha and -gamma signatures in EMPPs after infection (Fig.1f).

4. Il17ra, Il1r1 and Crlf2 were only assessed at the mRNA level. How about their protein levels? Are they expressed by all EMPPs and BEMPs but at a different level, or just a subset of these cells. What are the effects of IL-25 or TSLP injection on EMPPs and BEMPs?

Response:

We found that the available antibodies for these receptors do not reliably detect receptor expression on progenitor populations. We therefore used microfluidics-based single cell qRT-PCR to estimate the proportion of Il1r1-expressing cells. We detected expression of Il1r1 in 121 of 127 EMPPs (95%), 79/93 BEMPs (85%) and

87/99 preMegEs (88%). Given that some level of dropout is likely for any single cell PCR approach this would indicate near-ubiquitous expression. The expression of *Il1r1* in preMegEs, BEMPs and EMPPs is shown below – as can be seen, while expression is detected in nearly all cells, expression in BEMPs is on average higher, and in preMegEs lower, than in EMPPs. We are happy to add this analysis to the manuscript if deemed important.

As for the effects of IL-25 and TSLP on these populations, this is an interesting question, but not really within the scope of our study, and we therefore did not pursue this avenue of investigation.

Fig.R2. Expression of *Il1r1* in preMegEs, EMPPs and BEMPs, as measured by microfluidics-based qRT-PCR. The expression level is normalised to housekeeping gene expression (geometric mean of *B2m* and *Hprt*). P-values for comparison of the 3 populations were calculated using the Wilcoxon rank sum test.

[REDACTED]

6. GATA2 haploinsufficiency has been reported. Any comments on why *Gata2*^{+/-} mice do not have any phenotype?

Response:

Haploinsufficiency typically occurs for very GATA2 strong loss-of-function alleles (loss of expression or ZnF2 mutation leading to loss of DNA binding) – neither of these are affected by the G320D mutation. We do see milder phenotypes in the heterozygous genotype which sometimes reach statistical significance (see e.g. Fig.6b, Ext. Data Fig.8f,g). Generally speaking, the *D*^{+/-} phenotype is intermediate between *+/+* and *D/D*, as would be expected.